# The hexosamine biosynthetic pathway rescues lysosomal dysfunction in Parkinson's disease patient iPSC derived midbrain neurons

Willayat Y. Wani[1], Friederike Zunke[1,2], Nandkishore R. Belur[1] & Joseph R. Mazzulli [1] ✉

Disrupted glucose metabolism and protein misfolding are key characteristics of age-related neurodegenerative disorders including Parkinson's disease, however their mechanistic linkage is largely unexplored. The hexosamine biosynthetic pathway utilizes glucose and uridine-5'-triphosphate to generate N-linked glycans required for protein folding in the endoplasmic reticulum. Here we find that Parkinson's patient midbrain cultures accumulate glucose and uridine-5'-triphosphate, while N-glycan synthesis rates are reduced. Impaired glucose flux occurred by selective reduction of the rate-limiting enzyme, GFPT2, through disrupted signaling between the unfolded protein response and the hexosamine pathway. Failure of the unfolded protein response and reduced N-glycosylation caused immature lysosomal hydrolases to misfold and accumulate, while accelerating glucose flux through the hexosamine pathway rescued hydrolase function and reduced pathological α-synuclein. Our data indicate that the hexosamine pathway integrates glucose metabolism with lysosomal activity, and its failure in Parkinson's disease occurs by uncoupling of the unfolded protein response-hexosamine pathway axis. These findings offer new methods to restore proteostasis by hexosamine pathway enhancement.

Deficits in protein quality control machinery occur during the aging process and are thought to play a major role in the etiology of chronic neurodegenerative diseases[1]. The accumulation of insoluble protein aggregates is a cardinal feature of many age-related neurodegenerative diseases and directly implicate disruptions in protein homeostasis (or proteostasis) pathways[2]. Parkinson's disease (PD) is characterized by aberrant aggregation of multiple proteins[3], including α-synuclein (α-syn)-containing Lewy bodies and Lewy neurites that histopathologically define the disease[4]. Genetic studies also implicate lysosomal components as key contributors to PD pathogenesis[5–7]. For example, loss-of-function variants in *GBA1* that encodes β-glucocerebrosidase

(GCase) are among the strongest risk factors for PD[8]. Previous work from our group and others showed that reduced GCase activity results in lysosomal dysfunction and initiates pathogenic α-syn aggregation[9–11]. Multiple studies have shown that GCase activity is also reduced in PD patients that express wild-type *GBA1*[9,12], suggesting that loss of GCase function plays a common role in sporadic synucleinopathies. We previously showed that α-syn accumulation disrupts the trafficking and activity of wild-type GCase and other lysosomal hydrolases, creating a pathogenic self-propagating cycle of α-syn accumulation[9,13–15]. Several groups have shown that α-syn disrupts ER-Golgi trafficking in different PD models[16–19], and we recently found that

[1]The Ken and Ruth Davee Department of Neurology, Northwestern University Feinberg School of Medicine, Chicago, IL 60611, USA. [2]Department of Molecular Neurology, University Hospital Erlangen, Friedrich-Alexander University Erlangen-Nürnberg, Erlangen, Germany. ✉e-mail: jmazzulli@northwestern.edu

α-syn directly interferes with ER function and the unfolded protein response (UPR) causing the misfolding and aggregation of GCase[20]. Despite the accumulation of aggregated proteins in the ER, we found that PD patient-derived midbrain cultures failed to upregulate XBP1, a transcription factor that is controlled by the inositol-requiring enzyme 1-α (IRE1α) branch of the UPR and upregulates chaperones GRP78 and GRP94 during proteomic stress[20]. Collectively, these studies indicate that α-syn interferes with multiple branches of the proteostasis network, however our understanding of the mechanisms that contribute to the chronic accumulation of protein aggregates is incomplete.

Parallel to an age-related decline in proteostasis, glucose utilization in the brain declines during normal aging[21]. Dramatic changes in glucose utilization occur in the brains of living PD patients and correlate with the severity of clinical symptoms[22–25]. Cognitive dysfunction often occurs late in the clinical course of PD, and altered glucose metabolism can be observed in cortical regions of non-demented PD patients[22], suggesting it may play an early causal role in neurodegeneration. In addition to imaging studies, gene expression microarray studies of PD postmortem tissue have identified reduced expression of gene networks involving glucose metabolism and other bioenergetic pathways in the earliest stages of disease, suggesting that reduced glucose utilization precedes neuronal death[26]. Studies in animal models show that enhancing glucose flux protects against dopaminergic cell death and reduces α-syn pathology, although the mechanisms are not completely understood[27]. These data suggest a close association between perturbed glucose metabolism and PD pathogenesis. However, the mechanistic connection between impaired glucose flux and proteostasis failure is unclear.

Glucose flux through the hexosamine biosynthetic pathway (HBP) is essential for generating the high-energy sugar-nucleotide donor uridine-diphosphate-N-acetylglucosamine (UDP-GlcNAc), which provides the main glycan source for protein N-glycosylation[28]. The HBP begins with the conversion of obligatory intermediate fructose-6-phosphate (F-6-P) from the initial steps of glycolysis, and glutamine to form glucosamine-6-phosphate by the rate-limiting enzyme, glutamine:F-6-P transaminase (GFPT), ultimately forming UDP-GlcNAc. Two isozymes of GFPT exist (GFPT1 and GFPT2) and are expressed from distinct genes with variable expression patterns. In the central nervous system, GFPT2 (GFAT2, EC 2.6.1.16) is the predominant isoform[29]. UDP-GlcNAc is utilized in the committed step of N-glycosylation by dolichyl-phosphate-N-acetyl-glucosamine-phosphotransferase-1 (DPAGT1), which is the tunicamycin target[30,31]. The addition of N-glycans is required for the proper folding and trafficking of proteins that mature through the ER-Golgi pathway including lysosomal hydrolases[32]. In vivo studies have shown that enhancing glycosylation through activation of the HBP can improve proteostasis and extend lifespan in worms[33,34]. The importance of the HBP and N-glycosylation in proteostasis is also demonstrated by genetic diseases caused by variants in key HBP enzymes. For example, a rare congenital disorder of glycosylation caused by loss-of-function variants in DPATG1 primarily affects the nervous system and is clinically characterized by hypokinesia, cognitive impairment, and microcephaly[35–37]. Here, we explored the link between glucose utilization by the HBP and proteostasis failure in PD. We found that the HBP integrates glucose metabolism with lysosomal function, and that impaired glucose flux through the HBP is a causative factor in driving proteostasis failure in PD.

## Results

### Reduced N-glycosylated proteins in PD models and DLB brain occurs through deficits in N-glycan synthesis

To assess changes in N-glycosylation, we measured the total levels of protein N-glycans in PD midbrain cultures derived from iPSC lines (iPSn) that express the disease-causing A53T α-syn pathogenic variant[38]. Previous analysis of this model showed that pathological

α-syn aggregates develop by 60 days in culture, followed by the accumulation of immature GCase and lysosomal dysfunction at day 75, then neurodegeneration after day 100[13]. We directly measured N-glycosylated proteins using concanavalin A (Con A) conjugated to biotin, a lectin that is specific for N-glycans. The specificity of Con-A for binding N-glycans was confirmed by digestion of cell line extracts (H4 neuroglioma) with peptide-N-glycosidase F (PNGase) that removes N-linked oligosaccharides from glycoproteins. Western blot analysis using Con A-biotin indicated a reduction in the N-glycosylated proteins in PNGase-treated lysates indicating specific detection of N-glycans (Fig. S1A). Analysis of protein lysates from A53T iPSn with Con A-biotin indicated a dramatic decline in N-glycosylated proteins at day 90 compared to isogenic corrected controls (Corr) (Fig. 1A). Although treatment of Corr iPSn with tunicamycin reduced N-glycans by ~50% as expected, tunicamycin had no effect on A53T iPSn (Fig. 1A). This suggests that defects in N-glycosylation occur within the ER at the level of the tunicamycin target, DPAGT1, or on upstream HBP enzymes involved in the synthesis of N-glycan precursors.

We next determined if N-glycosylation was perturbed in vivo by analyzing brain lysates from A53T α-syn transgenic mice[39]. A53T mice develop insoluble α-syn inclusions throughout the neuraxis and neurological dysfunction that occurs in an age-dependent manner[39,40]. We found that N-glycosylated proteins were reduced by ~40% in brains from symptomatic A53T mice compared to age-matched non-transgenic controls (Fig. 1B). To assess the relevance of these findings to human disease, we measured N-glycosylated proteins in lysates from postmortem frontal cortical samples of synucleinopathy patients including Dementia with Lewy bodies (DLB) and Brainstem Lewy Body disease (BLBD). Analysis of other diseases including Alzheimer's disease (AD) and Progressive Supranuclear Palsy (PSP) was also done to assess specificity. We only found a significant reduction in N-glycosylated proteins of DLB brains compared to control, although mean values for other disease groups showed a subtle reduction but without significance (Fig. 1C). Taken together, these results indicate that global N-glycosylation detected by Con-A detects a more dramatic reduction in synucleinopathies compared to other neurodegenerative diseases assessed here.

To gain insight into the mechanism of N-glycan reduction, we measured the rate of N-glycan synthesis in PD culture models. The incorporation of oligosaccharides on proteins was examined by pulse-labeling living cultures with an azide-tagged mannose, N-azidoacetylmannosamine-tetraacylated (Ac4-ManNAz). Specific detection of proteins that incorporate Ac4-ManNAz was achieved through a Staudinger reaction with phosphine-biotin, followed by detection with streptavidin[41]. We first established the specificity of Ac4-ManNAz labeling in differentiated human neuroblastoma (SH-SY5Y) cells by digesting protein lysates with PNGase, which resulted in a ~50% reduction in Ac4-ManNAz signal (Fig. S1B). We next determined the levels of incorporated N-glycans in midbrain cultures and found a reduction at 18 and 36 h (h) post-labeling in A53T iPSn (Fig. 1D). Quantification of global protein translation showed that A53T iPSn were not different than controls, indicating that the reduction of incorporated N-glycans was not a reflection of changes in protein synthesis (Fig. S1C). These data suggest that reduced N-glycosylated protein occurs through a decline in the rate of N-glycan synthesis in the ER of A53T iPSn.

Since we previously found that immature, non-glycosylated lysosomal hydrolases accumulate and aggregate in PD iPSn[9,13,20], we next measured the amount of N-glycosylation that occurs specifically on lysosomal hydrolases. We first measured the beta subunit of hexosaminidase that comprises the B isozyme (Hex B), since its glycosylation and maturation can be easily followed by molecular weight changes on SDS-PAGE gels[42]. Treatment of control cells with tunicamycin blocked Hex B glycosylation, indicated by a reduction in the ~64 kDa glycosylated form and a concomitant increase in the 55 and

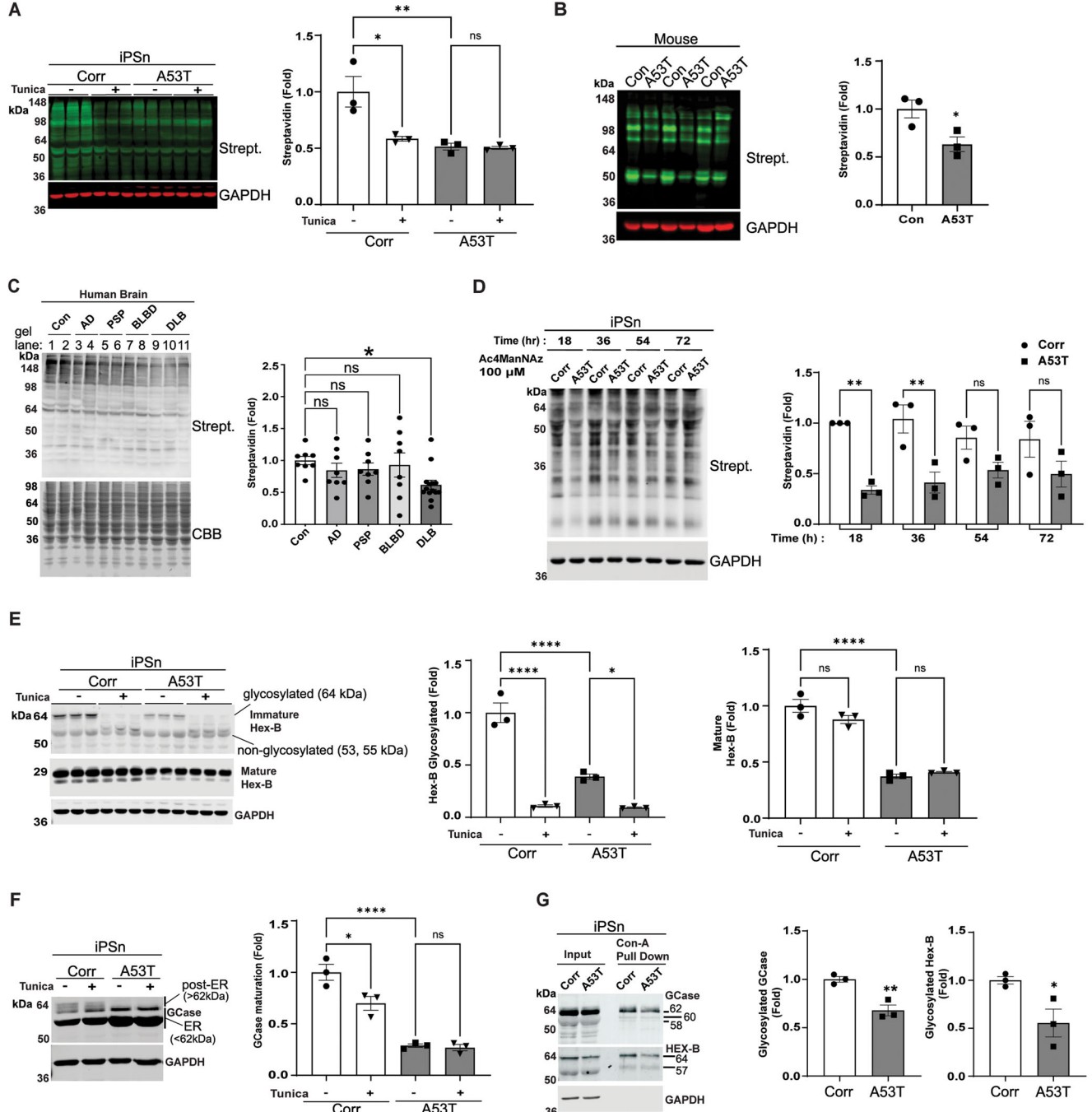

**Fig. 1 | Cellular and in vivo synucleinopathy models exhibit decreased N-glycosylated proteins.** Western blot analysis of N-glycosylated proteins in **A** A53T or isogenic corrected (Corr) iPSn at day 90 in the presence or absence of Tunicamycin (Tunica) for 24 h. Streptavidin (Strept.). **B** Cerebellar brain lysate from symptomatic A53T α-syn transgenic and age-matched control (Con) mice in the age range of 10–14 months, using biotinylated Con-A, detected by IRDye labeled streptavidin antibody. GAPDH is a loading control. Quantification is shown on the right (*n* = 3). **C** Post-mortem cortical human brain samples from Control (Con), Alzheimer's Disease (AD), Progressive Supranuclear Palsy (PSP), brainstem Lewy Body disease (BLBD), and Dementia with Lewy bodies (DLB), using biotinylated Con-A, detected by IRDye labeled streptavidin antibody (For DLB *n* = 14 and *n* = 8 for all other groups). CBB is used as loading control. Quantification is shown on the right. **D** N-glycan synthesis was measured by metabolic labeling of Corr and A53T iPSn with Ac4ManNAz, followed by cell lysis at 18, 36, 54, and 72 h. Labeled glycans were conjugated to biotin using biotin-phosphine. Biotinylated proteins were analyzed by Western blot using IRDye labeled streptavidin antibody. GAPDH is a loading control (*n* = 3). **E** Western blot analysis of Hex B in A53T or Corr iPSn at day

90 in the presence or absence of Tunicamycin (Tunica). GAPDH was used as a loading control. Quantification of glycosylated and mature forms of Hex-B is shown on the right of blot (*n* = 3). **F** Western blot analysis of GCase in A53T or Corr iPSn at day 90 in the presence or absence of Tunica. GAPDH was used as a loading control. Quantification of GCase maturation is shown on the right of blot (*n* = 3). ER endoplasmic reticulum. **G** Western blot of analysis of N-glycosylated GCase and Hex-B from Corr and A53T iPSn. N-glycosylated proteins were precipitated using Con-A conjugated to biotin, and streptavidin agarose beads were used to collect precipitated N-glycosylated proteins. Quantifications are shown on the right and normalized to input (*n* = 3). Scatter plots represent individual culture well replicates, individual mouse brains, or individual human brain samples. For all quantifications, values are the mean ± SEM, *$p < 0.05$, **$p < 0.01$, ***$p < 0.001$, ****$p < 0.0001$. ANOVA with Tukey's post hoc test was used for (**A**, **E**, **F**). ANOVA with Dunnett's test was used for (**C**), ANOVA with Šídák's multiple comparisons test was used for (**D**) and Student's two-sided *t*-test was used for (**B**, **G**). Source data are provided as a Source Data file.

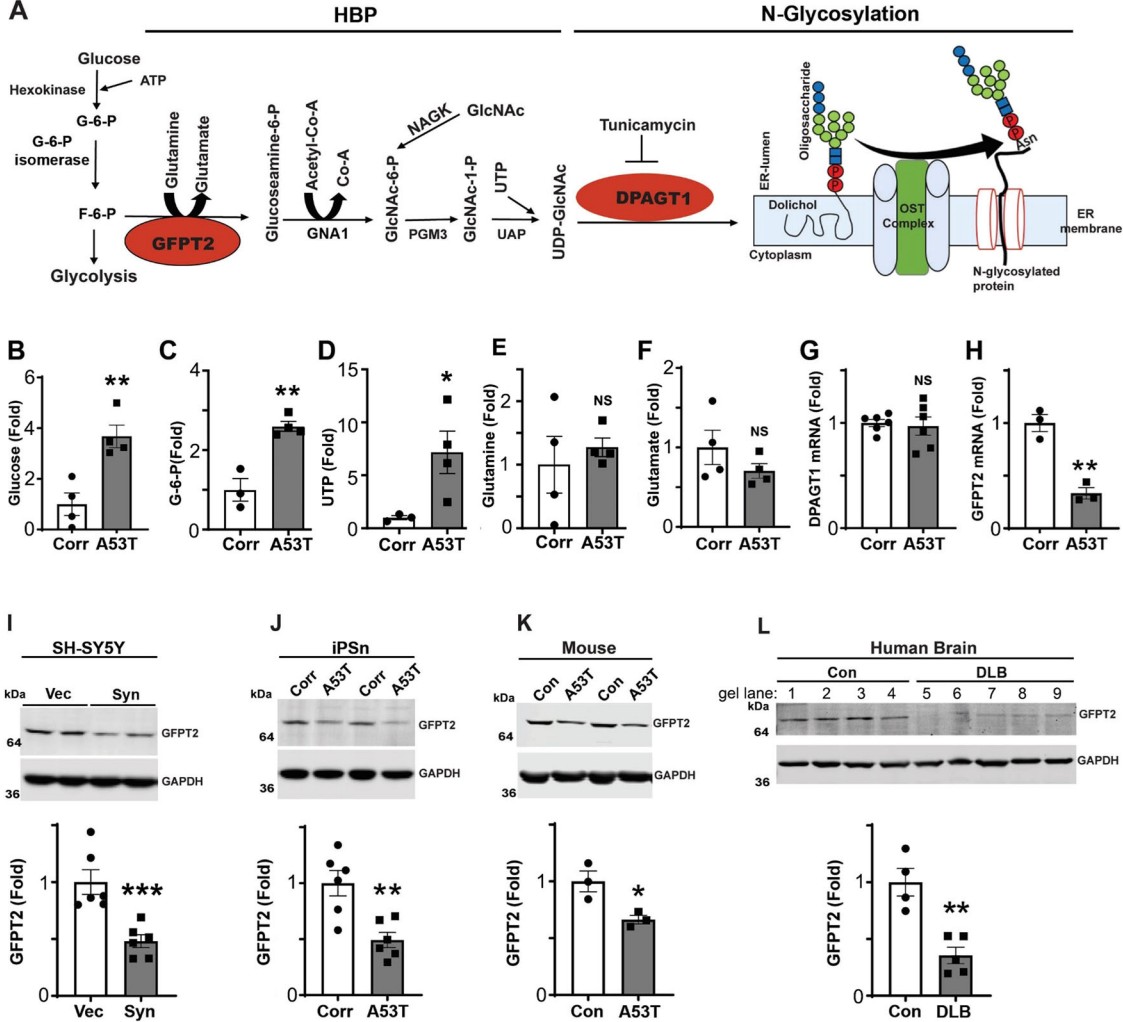

**Fig. 2 | Hexosamine-biosynthetic pathway (HBP) is disrupted in synucleino-pathy models and DLB patient brain. A** Schematic showing HBP and N-glycosylation pathway. UDP-GlcNAc Uridine-diphosphate-N-acetylglucosamine, F-6-P Fructose-6-phosphate, HK-1 Hexokinase-1, G-6-P glucose-6-phoshate, GFPT2 glutamine: F-6-P transaminase-2, GNA1 GlcN-6P acetyltransferase, acetyl-CoA acetyl coenzyme A, PGM3 GlcNAc phosphomutase, UAP1 UDP-GlcNAc pyropho-sphorylase, NAGK N-acetylglucosamine kinase, OST oligosaccharyltransferase, **B–F** Targeted metabolomic analysis of A53T or Corr iPSn at day 90 by Liquid Chromatography-Mass Spectrometry LC/MS (**B**) Glucose (*n* = 4) (**C**) Glucose-6-Phosphate (Corr *n* = 3; A53T *n* = 4) (**D**) Uridine triphosphate (UTP) (Corr *n* = 3; A53T *n* = 4) (**E**) Glutamine (*n* = 4) (**F**) Glutamate (*n* = 4). **G** Quantitative RT-PCR analysis of

DPAGT1 (*n* = 6), NS not significant. **H** Quantitative RT-PCR analysis of GFPT2 mRNA in A53T or Corr iPSn at day 90 (*n* = 3). **I–L** Western blot analysis of GFPT2 in (**I**) SH-SY5Y vector (vec) and stable line overexpressing WT α-syn (Syn) (*n* = 6), (**J**) iPSn (Corr and A53T) (*n* = 6) (**K**) Non-transgenic control (Con) and A53T transgenic mouse brain (cerebellum) (*n* = 3) and (**L**) post-mortem cortical human brain sam-ples from Control (Con) and Dementia with Lewy body patients (DLB). GAPDH was used as loading control. Quantifications are shown on the right (con = 4; DLB *n* = 5). Scatter plots represent individual culture well replicates, individual mouse brains, or individual human brain samples. For all quantifications, values are the mean ± SEM, *\*p* < 0.05, *\*\*p* < 0.01, *\*\*\*p* < 0.001, *\*\*\*\*p* < 0.0001. Student's two-sided *t*-test was used for all comparisons. Source data are provided as a Source Data file.

53 kDa non-glycosylated forms in isogenic control iPSn (Fig. 1E). Ana-lysis of untreated A53T iPSn showed a 60% decrease in the levels of the 64 kDa glycosylated Hex B compared to controls, and a reduced response to tunicamycin (Fig. 1E). Similarly, tunicamycin treatment reduced GCase maturation in Corr iPSn by ~30%, indicated by increased immature (55 and 62 kDa) and reduced mature (>62 kDa) forms of GCase (Fig. 1F). Untreated A53T iPSn exhibited a dramatic decline in mature GCase compared to Corr iPSn and no response to tunicamycin (Fig. 1F). We confirmed that the N-glycosylation of hydrolases was reduced by precipitating proteins from iPSn lysates with Con A-biotin, followed by immunoblotting for GCase and Hex-B. This revealed a reduction in the amount of N-glycosylated ER forms of GCase migrating at 62, 60, and 58 kDa and Hex-B forms migrating at 64 kDa in A53T iPSn compared to Corr (Fig. 1G). These data indicate that reduced N-glycosylation of GCase and Hex B occurs in A53T iPSn, compromising hydrolase folding and maturation.

## Metabolomic analysis indicates disruptions in the HBP in PD iPSn

We next considered the possibility that reduced N-glycan synthesis was a result of low intracellular glucose or impaired turnover within the initial steps of glycolysis that provide essential HBP substrates (Fig. 2A). We performed targeted metabolomics of A53T iPSn to spe-cifically test this hypothesis. Instead of glucose depletion, we found a ~4-fold accumulation of intracellular glucose and glucose-6-phosphate (G-6-P), and no change was observed in hexokinase protein levels (Fig. 2B, C; Fig. S2A). No change was also found in Glyceraldehyde 3-phosphate dehydrogenase (GAPDH), which functions in glycolytic processing downstream of the HBP shunt (Fig. S2A, right). Further-more, uridine 5′-triphosphate (UTP), which is utilized downstream in the HBP by UDP-N-acetylhexosamine pyrophosphorylase (UAP1) to generate UPD-GlcNAc, was also elevated by ~7 fold (Fig. 2D). No change was found in other HBP metabolites including glutamine or glutamate

(Fig. 2E, F), which may be due to their involvement in multiple metabolic pathways beyond the HBP. Next, we measured the expression levels of the most critical enzymes in the HBP including the rate-limiting GFPT2 and DPAGT1. While the mRNA of DPAGT1 was not changed, GFPT2 mRNA was dramatically reduced in A53T iPSn compared to isogenic controls (Fig. 2G, H). We confirmed that GFPT2 protein was reduced by western blot analysis, a finding that was reproducible in all α-syn accumulation models examined, including SH-SY5Y cells, A53T iPSn, and A53T transgenic mice (Fig. 2I–K). GFPT2 reduction was also observed in a separate PD iPSn line that was previously characterized and accumulates pathological α-syn through a triplication in the *SNCA* genomic region (*SNCA*-3x)[20] (Fig. S2B). Assessment of DLB patient brain showed a ~70% reduction in GFPT2 protein (Fig. 2L). Measurement of DPAGT1 protein by western blot showed a 1.6-fold elevation in A53T iPSn (Fig S2C) while other enzymes in the HBP, including glucosamine 6-phosphate N-acetyltransferase (GNA1) and N-acetyl-D-glucosamine kinase (NAGK), showed no change (Fig. S2D, E). We measured the levels of a critical component of oligosaccharyltransferase complex, OST48, which catalyzes the en-bloc transfer of the dolichol-linked oligosaccharide onto proteins in the ER[43]. Western blot analysis indicated no change in OST48 in A53T iPSn (Fig. S2F) suggesting that N-glycosylation downstream of DPAGT1 is likely not perturbed in PD iPSn. Together, these data suggest that reduced N-glycan synthesis in PD iPSn could be due to depleted GFPT2 mRNA and protein.

## GFPT2 responds to protein misfolding stress through XBP1 and is disrupted in PD patient cultures

We next sought to determine the mechanism of GFPT2 reduction. Since GFPT2 mRNA was reduced in A53T iPSn, we hypothesized that transcriptional regulation could be perturbed. Previous studies showed that GFAT1, the non-neuronal isozyme of GFPT2 that is expressed on a separate gene, is transcriptionally controlled by the unfolded protein response (UPR)[33]. Additionally, our previous work showed that α-syn impedes the activation of the UPR pathway in response to misfolded proteins[20], providing a potential explanation for reduced GFPT2 mRNA. Since transcriptional control of GFPT2 by the UPR has not been examined previously in human neurons, we first tested if tunicamycin triggers the upregulation of GFPT2 in Corr iPSn. We found that while tunicamycin induced a ~ 2-fold upregulation of GFPT2 mRNA in Corr iPSn, A53T iPSn showed a significantly reduced response to tunicamycin compared to controls (Fig. 3A). These findings were confirmed by western blot, which showed that tunicamycin increased GFPT2 protein in Corr iPSn, but had no effect on A53T iPSn (Fig. 3B). We validated this result in a distinct *SNCA*-3x iPSn with matching isogenic corrected control (Fig. S3). Examination of other UPR-responsive transcripts, including XBP1 and its downstream transcriptional target GRP78 also revealed that A53T iPSn were less responsive to tunicamycin (Fig. 3B–D). These data suggest that GFPT2 expression can be triggered by the UPR in human iPSC-derived midbrain neurons and this pathway is compromised by α-syn accumulation, consistent with previous studies[20].

ER stressors including tunicamycin induce oligomerization and autophosphorylation of IRE1, which activates its endoribonuclease domain to excise a 26-nucleotide fragment from unspliced Xbp1 mRNA. This forms a spliced, activated transcription factor called XBP1s[44]. XBP1s in turn transactivates genes that enhance the protein-folding capacity of the ER including GRP78 and GRP94[44]. Since previous studies found that XBP1s activates GFPT1[33], we next ascertained whether GFPT2 transcription is also regulated through XBP1s under stress conditions. We disabled IRE1α in Corr iPSn using the selective pharmacological inhibitor MKC8866[45], in the presence or absence of tunicamycin. Tunicamycin induced the upregulation of both XBP1s and GFPT2 mRNA, however, this effect was abrogated by IRE1α inhibition (Fig. 3E, F). These results indicate that IRE1α-XBP1s signaling

regulates the HBP pathway under ER stress through the transcriptional regulation of GFPT2.

Our previous work indicated that misfolded GCase in the ER does not trigger the UPR in PD iPSn[20]. To determine if specific stress induced by protein-misfolding can trigger the XBP1-GFPT2 pathway, we expressed an unstable form of GCase that causes Gaucher disease and induces ER stress[20], L444P, in both Corr and PD (*SNCA*-3x) iPSn. While expression of L444P GCase increased the levels of XBP1s and GFPT2 mRNA in Corr iPSn, PD iPSn showed no response (Fig. 3G, H). These data indicate that GFPT2 reduction in PD iPSn may occur from ER perturbations that result in a deficiency of sensing or transducing protein misfolding signals.

Since GFPT2 expression is upregulated by XBP1s, we next attempted to rescue A53T iPSn by directly expressing XBP1s using lentivirus. This resulted in a significant elevation of GFPT2 mRNA, but also other downstream UPR targets including GRP78 and GRP94 (Fig. 4A). XBP1s overexpression increased GFPT2 protein by over 50%, which was sufficient to achieve a downstream increase in N-glycosylated proteins (Fig. 4B, C). Western blot showed that XBP1s also restored the expression of other UPR proteins including GRP94 and GRP78 (Fig. 4B, C). Consistent with improved ER proteostasis, XBP1s overexpression increased GCase enzyme activity in A53T iPSn by ~70% (Fig. 4D), which was sufficient to reduce α-syn levels by ~50% (Fig. 4E). Together, these data suggest that GFPT2 reduction and ER proteostasis failure in PD occurs upstream of XBP1s.

## Genetic enhancement of the HBP rescues lysosomal dysfunction and reduces pathogenic α-syn

To directly examine the relationship between the HBP and lysosomal function in PD, we next determined if expressing key proteins involved in N-glycosylation could rescue downstream lysosomal phenotypes. First, we attempted to rescue lysosomal dysfunction in A53T iPSn through overexpression of DPAGT1. Lentiviral-mediated transduction and western blot of iPSn confirmed DPAGT1 overexpression (Fig. S4A), and increased activity of the construct was seen through measuring N-glycosylation of proteins in transfected HEK cells (Fig. S4B). However, transduction of A53T iPSn with DPAGT1 lentivirus had no effect on the levels of N-glycosylated proteins, lysosomal hydrolase maturation, or pathological α-syn (Fig. S4C–E). These data are consistent with the hypothesis that reduced N-glycosylation in synucleinopathy occurs upstream of DPATG1, within the HBP.

Since we found a reduction in GFPT2 protein levels (Fig. 2H–L), we next tested if increased GFPT2 expression could rescue lysosomal phenotypes in A53T iPSn. Cells were transduced with lentivirus expressing GFPT2 and western blot confirmed that GFPT2 protein levels were increased in A53T iPSn compared to those infected with a control GFP-expressing lentivirus (Fig. 5A). Analysis of total intracellular glucose levels showed a 10% reduction upon GFPT2 overexpression, which translated to an increase of N-glycosylated proteins by ~25% (Fig. 5B, C). GFPT2 increased the N-glycan binding activity of calnexin shown by Con-A pulldown/western blot, suggesting that chaperone function was improved by enhancing the HBP (Fig. 5D). In the same complexes, we found increased levels of N-glycosylated GCase by Con-A pulldown/western blot in GFPT2 infected cultures (Fig. 5D). We next tested if GFPT2 improved lysosomal hydrolase maturation by western blot analysis. We found a significant increase in glycosylated GCase migrating at 62 kDa, and an elevation in the mature (post-ER) forms migrating above 62 kDa (Fig. 5E). The effect on maturation was confirmed by digesting lysates with endoglycosidase H (Endo H), an enzyme that selectively cleaves protein-linked glycans in the ER, but not glycans that are modified in the Golgi. This revealed an increase in the levels of Endo H-resistant GCase upon GFPT2 overexpression (Fig. 5E). We next assessed the effect of GFPT2 on Hex B maturation by western blot. Similar to GCase, GFPT2 overexpression increased mature Hex B chain (29 kDa) by nearly 50% (Fig. 5F). mRNA

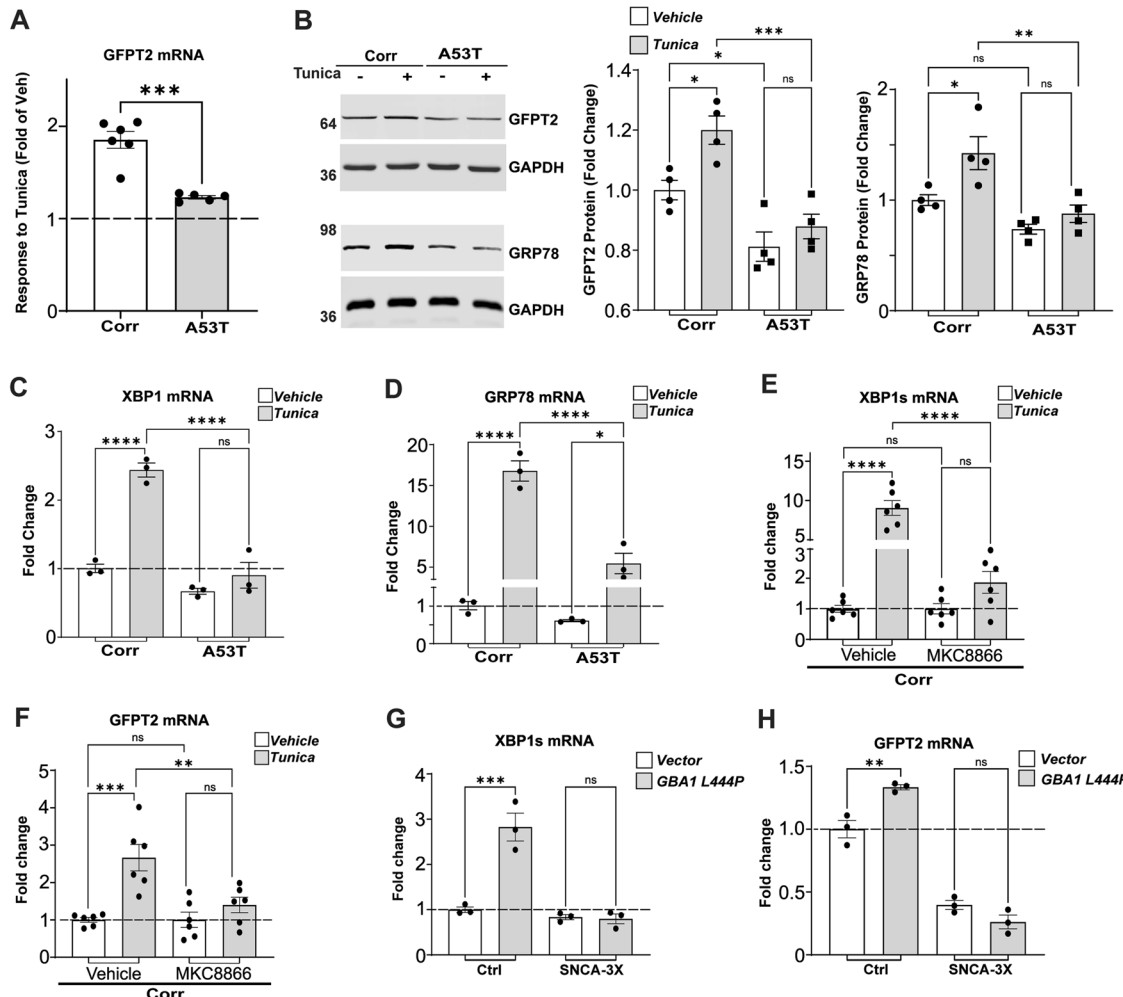

**Fig. 3 | GFPT2 is upregulated in response to tunicamycin and disrupted in PD iPSn. A** Quantitative RT-PCR analysis of GFPT2 mRNA in isogenic corrected (Corr) and A53T iPSn at day 90, after treatment with vehicle (DMSO) and tunicamycin (5 ug/ml) for 24 h (h) (Corr n = 6; A53T n = 5). **B** Western blot analysis of GFPT2 and GRP78 protein in iPSn (Corr and A53T) treated as in (**A**), with GAPDH as a loading control. Quantifications are shown on the right (*n* = 4). GFPT2 and GRP78 are from different blots, and GAPDH loading control shown below corresponds to the same matching blot. **C** Quantitative RT-PCR analysis of XBP1 mRNA in Corr and A53T iPSn at day 90, treated as in panel A (*n* = 3). **D** Quantitative RT-PCR analysis of GRP78 mRNA in Corr and A53T iPSn at day 90, treated as in (**A**) (*n* = 3).

**E** Quantitative RT-PCR analysis of XBP1s mRNA transcript in Corr iPSn at day 90 treated with 10 µM MKC8866 and tunicamycin (5 ug/ml) for 24 h (*n* = 6). **F** Quantitative RT-PCR analysis of GFPT2 mRNA transcript in Corr iPSn treated as in (**E**). **G** Quantitative RT-PCR analysis of XBP1s mRNA transcript in day 90 control and *SNCA*-3x (line 3x-2) iPSn infected to overexpress L444P *GBA1* (MOI = 1; dpi 14 days) (*n* = 3). **H** Quantitative RT-PCR analysis of GFPT2 mRNA transcript as in (**G**). Scatter plots represent individual culture well replicates. For all quantifications, values are the mean ± SEM, *p < 0.05, **p < 0.01, ***p < 0.001, ****p < 0.0001. Student's two-sided *t*-test was used for (**A**) and ANOVA with Tukey's post hoc test was used for (**B**–**H**). Source data are provided as a Source Data file.

quantification of GCase and Hex B showed no change by GFPT2 over-expression, suggesting that improved maturity occurred post-transcriptionally (Fig. 5G). Additionally, we found no change in the levels of GRP78, GRP94, or XBP1s mRNA or protein upon GFPT2 overexpression, suggesting that the effect on lysosomal hydrolase maturation was directly due to increased N-glycosylation as opposed to indirect effects (Figs. S5A, B). To determine whether GFPT2 could elevate lysosomal function, we performed a live-cell enzyme activity assay that assesses trafficking and maturation of GCase by discriminating between activity that occurs in lysosomal and non-lysosomal compartments of neurons[13,46]. Lysosomal GCase activity was significantly elevated by GFPT2, indicating that GCase targeting to lysosomal compartments was improved (Fig. 5H). Importantly, the increased activity was sufficient to reduce insoluble, pathological forms of α-syn in A53T iPSn shown by sequential extraction/western blot analysis (Fig. 5I). Finally, we tested if GFPT2 overexpression could improve neuron viability, using an established neurofilament immu-nostaining assay that reflects total neurite content in cultures.

GFPT2 significantly increased neurite content in A53T iPSn compared to vector controls, suggesting that neuron viability is improved (Fig. 5J).

To determine if GFPT2 overexpression could also rescue lysosomal function in *SNCA*-3x PD iPSn that accumulates wild-type α-syn, we infected *SNCA*-3x iPSn with lenti-GFPT2. Similar to A53T iPSn, GFPT2 increased calnexin binding activity toward N-glycans (Fig. S5C), and elevated lysosomal GCase activity while decreasing non-lysosomal activity, reflecting improved GCase trafficking to lysosomes (Fig. S5D, E). Reduced levels of pathological α-syn were also observed by GFPT2 (Fig. S5F). Taken together, these data show that hypo-N-glycosylation in synucleinopathy occurs through deficits in GFPT2, and restoring its expression can rescue lysosomal function, reduce α-syn, and improve neuronal health.

### Pharmacological enhancement of the HBP restores proteostasis in PD iPSn

Since we found that GFPT2 can rescue lysosomal function in PD iPSn, we next determined if N-glycan synthesis could be rescued by the

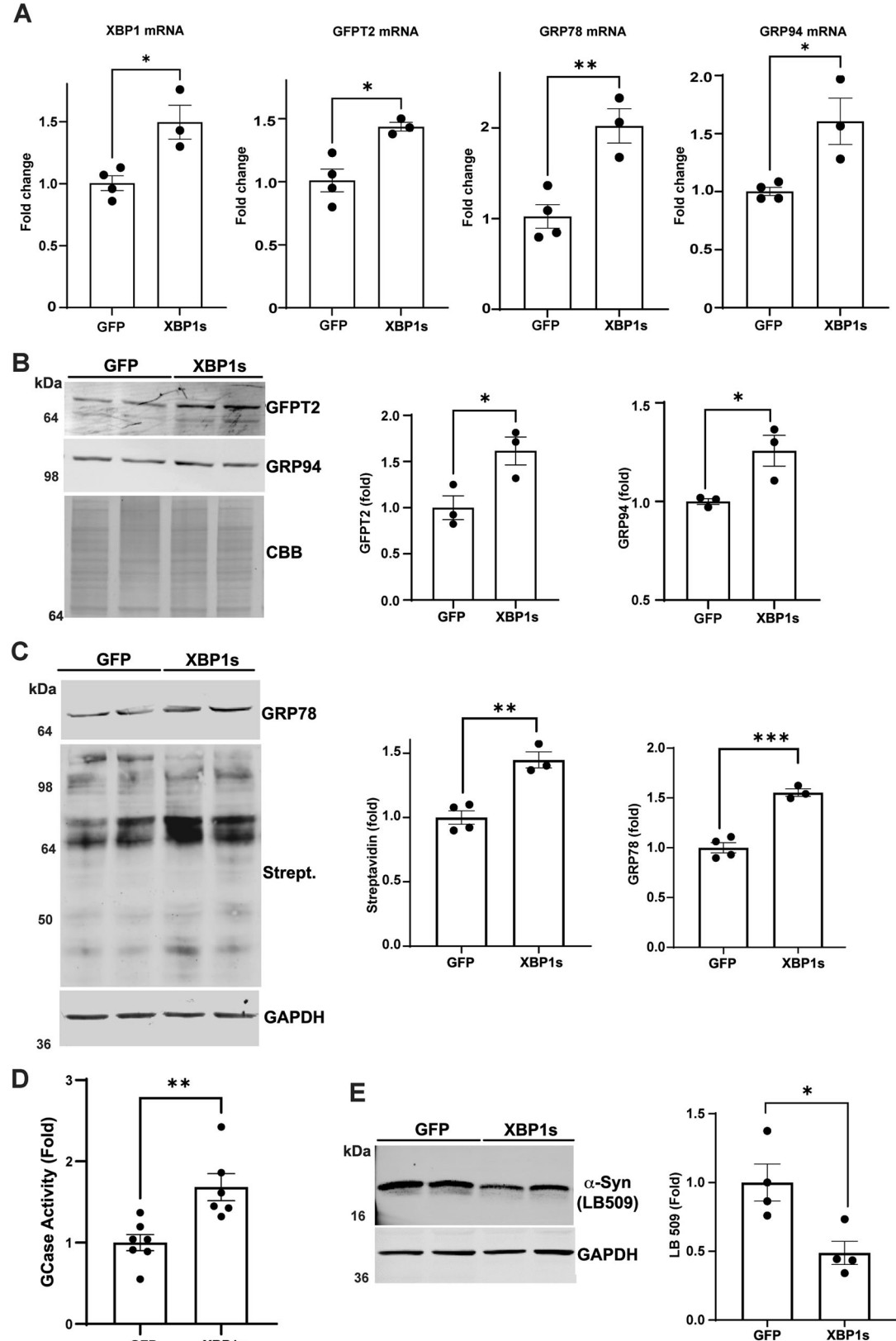

**Fig. 4 | XBP1s rescues the HBP and lysosomal function in PD iPSn.** A53T iPSn were infected with lentivirus expressing GFP (Control) or XBP1s at MOI of 1 at day 104 and analyzed at 16 days later, at day 120. **A** Quantitative RT-PCR analysis of XBP1, GFPT2, GRP78 and GRP94 in A53T iPSn at day 120 (GFP, *n* = 3; XBP1s, *n* = 4). **B** Western blot analysis of GRP94 and GFPT2 proteins in infected A53T iPSn, Coomassie blue (CBB) is used as a loading control (*n* = 3). **C** Western blot of GRP78 and N-glycosylated proteins using biotinylated Con-A, in A53T iPSn. GAPDH is a loading control. Quantification is shown on the right (GFP, *n* = 4; XBP1s, *n* = 3).

**D** GCase activity was measured in A53T iPSn. The activity was expressed relative to GFP infected iPSn as a control (GFP, *n* = 7; XBP1s, *n* = 6). **E** Western blot analysis of α-syn from Triton X-100-Soluble lysates using the LB509 antibody in A53T iPSn. GAPDH is a loading control. Quantification is shown on the right (*n* = 4). Scatter plots represent individual culture well replicates. For all quantifications, values are the mean ± SEM, *$p < 0.05$, **$p < 0.01$, ***$p < 0.001$, ****$p < 0.0001$. Student's two-sided *t*-test was used for all comparisons. Source data are provided as a Source Data file.

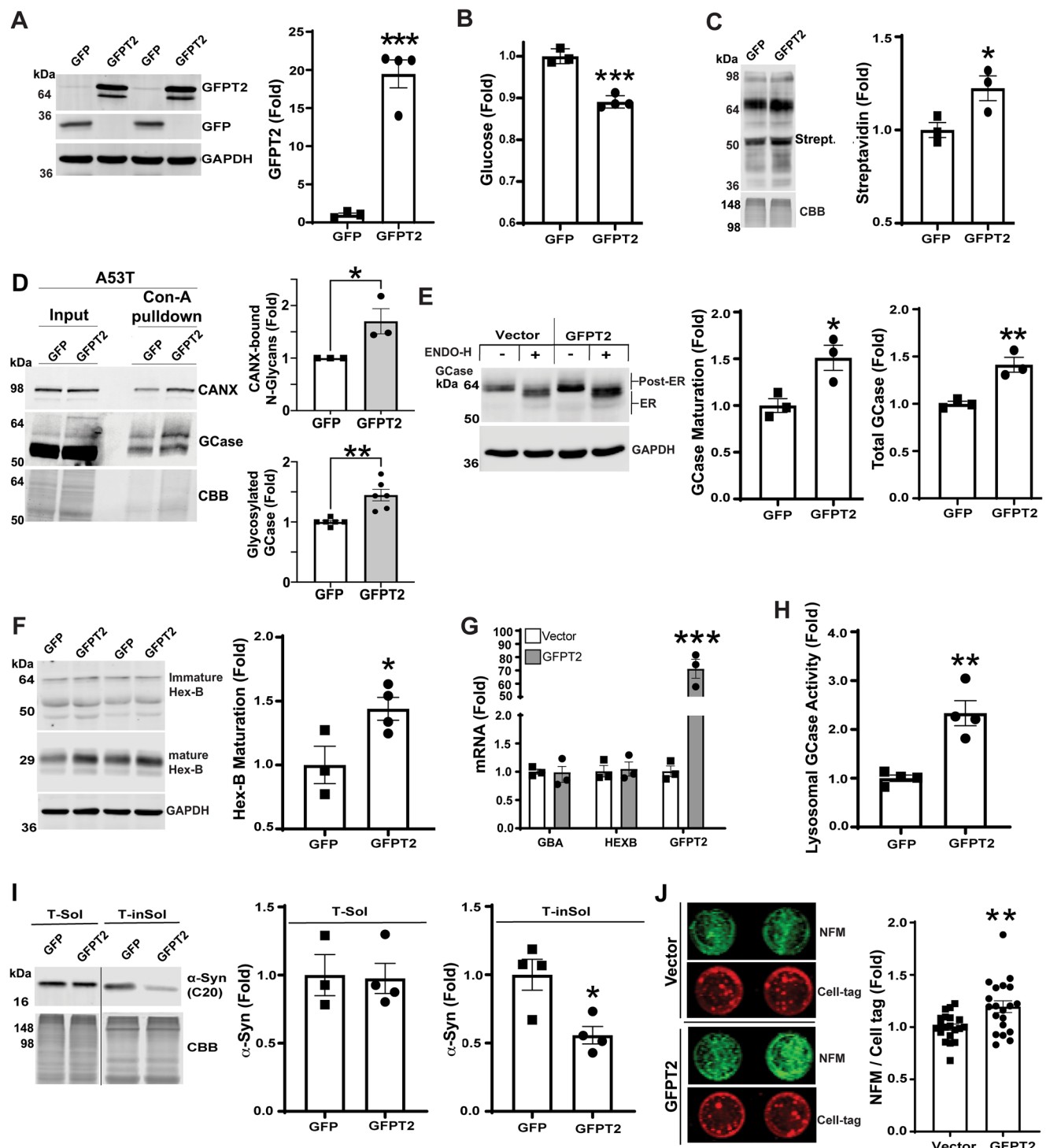

addition of HBP intermediates that act downstream of GFPT2. N-glycosylation can also be elevated through supplementation of N-acetylglucosamine (GlcNAc) by NAGK-mediated conversion into GlcNAc-6-P, which enters the HBP downstream of GFPT2 and ultimately forms UDP-GlcNAc (Fig. 2A). Previous studies have shown that exogenous GlcNAc is utilized by the HBP and can improve N-glycosylation in cancer models[47–49]. GlcNAc can also extend the lifespan of *C. elegans* through enhancing proteostasis[34]. Since we found that NAGK levels were not altered in A53T iPSn (Fig. S2E), we treated cultures with GlcNAc and quantified N-glycosylated proteins. GlcNAc treatment had little effect in Corr iPSn, but completely restored N-glycosylation in A53T iPSn and improved GCase solubility

as shown by sequential extraction/western blot analysis (Fig. 6A, B). GCase maturation increased by over 2-fold (Fig. 6C), while enzymatic GCase activity nearly doubled compared to vehicle controls (Fig. 6D). GlcNAc also rescued the N-glycosylation and maturation of Hex B in A53T iPSn but had no effect in isogenic control cells (Fig. 6E, F). Quantification of ER chaperones in A53T iPSn showed no changes by GlcNAc treatment, indicating that increased lysosomal hydrolase maturation and function occurred from increased N-glycosylation as opposed to indirect effect mediated by ER chaperone upregulation (Fig. S6A). GlcNAc treatment reduced pathological α-syn by ~75% and improved neuron viability (Fig. 6G–I). We confirmed these results in *SNCA*-3x iPSn, showing that GlcNAc treatment increased

**Fig. 5 | GFPT2 restores ER-Golgi trafficking of lysosomal hydrolases, GCase activity and reduces α-syn accumulation in PD iPSn. A** Western blot analysis of GFPT2 and GFP in day 90 A53T iPSn infected with lentivirus expressing GFP as a control, and GFPT2 at MOI-3. iPSn were analyzed 15 days after infection. GAPDH is a loading control. Quantification is shown on the right (GFP, $n = 3$; GFPT2, $n = 4$). **B** A53T iPSn were infected with lentivirus expressing GFP and GFPT2. Glucose levels were measured from lysates by enzymatic production of glucose-derived NADH in vitro followed by conversion of water-soluble tetrazolium (WST) to formazan, and normalized to total protein (GFP, $n = 3$; GFPT2, n = 4). **C** Western blot of N-glycosylated proteins using biotinylated Con-A, in A53T iPSn infected with lentivirus expressing GFP and GFPT2. GAPDH is a loading control. Quantification is shown on the right ($n = 3$). **D** Calnexin (CANX) binding activity was measured in PD iPSn (A53T) infected with GFP or GFPT2 as in (**A**). Quantification of N-glycan binding was done by Con-A pull-down/CANX western blot ($n = 3$ culture wells). Western blot for GCase was done on the same precipitates and quantified on the right ($n = 6$ culture wells). **E** GCase levels and maturation were measured by western blot analysis using endoglycosidase H (endo H) digestion of lysates from GFP and GFPT2 infected A53T iPSn. GAPDH is a loading control. $n = 3$, ±SEM, *$p < 0.05$. Total GCase levels were quantified from the non-endo H digested lanes. **F** Western blot analysis of Hex B in A53T iPSn infected with GFP and GFPT2, using GAPDH as a loading

control. Quantifications of mature Hex-B is shown on the right (GFP, $n = 3$; GFPT2, $n = 4$). Note that the Immature Hex B bands are shown at a higher intensity compared to the mature Hex B bands, to avoid saturation of the mature forms. **G** Quantitative RT-PCR analysis of *GBA1*, *HEXB,* and *GFPT2* in A53T iPSn at day 90, infected with control GFPT2 lentivirus at MOI-3, 15 days after infection ($n = 3$). **H** Analysis of lysosomal GCase activity in living A53T iPSn infected with empty vector or GFPT2 lentivirus at MOI-3. Fluorescent substrate degradation was evaluated in a microplate reader for 3 h. Activity within acidic cellular compartments was determined by quantifying the response to bafilomycin A1 (Baf A1) ($n = 4$). **I** Western blot analysis of α-syn from Triton X-100 soluble (T-sol) (GFP, $n = 3$; GFPT2 $n = 4$) and insoluble (T-insol) ($n = 4$) fractions using the C20 antibody, in A53T iPSn infected with vector and GFPT2 lentivirus. Coomassie brilliant blue (CBB) is a loading control. Quantification is shown on the right. The line indicates that the T-sol and T-insol samples were run on different gels/blots. (**J**) Neuron viability was assessed in A53T iPSn infected with vector and GFPT2 lentivirus, by quantification of neurofilament (NFM) content and normalized to a general cell volume stain (cell-tag). Quantification is shown on the right ($n = 20$). Scatter plots represent individual culture well replicates. For all quantifications, values are the mean ± SEM, *$p < 0.05$, **$p < 0.01$, ***$p < 0.001$, Student's two-sided $t$-test was used for all comparisons. Source data are provided as a Source Data file.

N-glycosylated proteins and reduced α-syn levels (Fig. S6B, C, vehicle conditions). Furthermore, treatment with the pan-cathepsin inhibitor Leupeptin, partially blocked α-syn clearance, indicating that GlcNAc-mediated α-syn reduction occurs through lysosomes (Fig. S6C). Together, these results show that HBP can be enhanced by GlcNAc supplementation in PD iPSn, resulting in rescued lysosomal phenotypes and reduced α-syn.

## Discussion

Although changes in glucose metabolism and protein aggregation are two well-established features of age-related neurodegenerative disorders, their mechanistic connection was not fully understood. Reduced glucose utilization in the brain is associated with a decline in neural function and disease severity in PD patients, but it was previously unclear whether these changes are a cause or consequence of disease. We find that a decline in the glucose utilization through the HBP leads to proteostasis failure, resulting in lysosomal dysfunction and protein aggregation. Improving the HBP through GFPT2 or GlcNAc can restore lysosomal function, demonstrating a direct relationship between glucose flux and proteostasis in PD. Impaired glucose utilization is evident in PD iPSn by the accumulation of intracellular glucose, G-6-P, and UTP that is utilized by the HBP to generate precursors required for N-glycosylation. The HBP is an off-shoot of glycolysis that normally utilizes a minor portion of the total intracellular glucose in non-neuronal cells[50], although the percentage of glucose utilized by the HPB in neurons has not been determined. It is possible that a chronic reduction of GFPT2 activity over time could result in a significant increase of glucose, as we observe in our long-term midbrain cultures aged for 90 days. Furthermore, the HBP can increase glucose flux even further, under periods of stress to maintain proteostasis, including activation of the unfolded protein response in the ER[33]. Under proteomic stress, it is likely that chronic GFPT2 reduction in PD iPSn contributes to increased intracellular glucose levels due to the lack of conversion into N-glycan precursors.

Our data indicates that under physiological conditions, the HBP integrates metabolic information with the UPR and lysosomes by altering intracellular glucose flux (summarized in Fig. 7). Alterations in glucose flux through the HBP directly influence protein folding, trafficking, and lysosomal clearance pathways to maintain proteostasis. In PD, this balance is severely perturbed by reduced glucose flux through the HBP. This was demonstrated by rescue experiments, where stimulating the HBP improved N-glycosylation, calnexin binding activity, lysosomal enzyme trafficking and activity, and reduced toxic α-syn aggregation (Figs. 4–6). Perturbed protein trafficking is thought to be a

key pathological process in PD[51], and our data suggest that enhancing the HBP can rescue this phenotype. Of critical importance is that HBP stimulation was able to improve the lysosomal activity of GCase, given its link to PD and DLB as an established genetic risk factor. Furthermore, reduced activity of wild-type GCase is a common feature of synucleinopathies[9]. Reduced GCase folding in the ER occurs as a result of GCase pathogenic variants as in the case of Gaucher disease, but also by α-syn accumulation. Our previous study indicated that wild-type GCase accumulates and forms aggregates in the ER compartment, thereby reducing lysosomal function[20]. Importantly, GlcNAc treatment improved GCase solubility and function in PD iPSn (Fig. 6B). Our findings offer a new method to enhance the folding and stability of lysosomal hydrolases, which could be used as future therapies to enhance lysosomal function and reduce toxic protein aggregates.

We found that reduced N-glycan synthesis occurred through a reduction in the rate limiting enzyme, GFPT2. The reduction of both mRNA and protein indicates that the effect is likely due to transcriptional reduction or reduced RNA stability, although more studies are required to delineate the mechanism. Previous studies showed that the HBP can be activated by the UPR through transcriptional upregulation of GFAT1[33]. Similarly, we found that triggering the UPR through tunicamycin treatment or expression of the highly unstable L444P GCase stimulated GFPT2 expression in controls, but not in PD iPSn (Fig. 3A, B, H). This is consistent with our previous work demonstrating that misfolded proteins fail to trigger an effective unfolded protein response in PD iPSn[20]. Using an IRE1 inhibitor, we found that XBP1s was required for GFPT2 upregulation (Fig. 3F). Furthermore, overexpression of the activated transcription factor, XBP1s, directly upregulated GFPT2 expression and rescued downstream N-glycosylation and lysosomal phenotypes in PD iPSn (Figs. 4, 7). Collectively our studies suggest that GFPT2 reduction in PD occurs through inefficient UPR transcriptional response when exposed to protein-misfolding stress, resulting in lysosomal dysfunction and augmented α-syn accumulation. Although more studies are required to delineate the upstream mechanisms for how PD iPSn fail to elicit the UPR under periods of stress, previous work suggests that impeded ER-Golgi trafficking of ATF6 may play a role[52]. ATF6 is a critical signal transducer and transcription factor of the UPR, which directly upregulates the expression of XBP1[44]. Upon sensing protein misfolding in the ER, membrane bound ATF6 is trafficked to the Golgi where it is cleaved and activated by site-1 and site-2 protease[53]. The N-terminal cytosolic fragment (ATF6(N)) is then imported into the nucleus where it activates the transcription of XBP1[44]. Since the XBP1 pathway fails to respond to protein-misfolding stress in PD iPSn[20] (Fig. 3C, G), and

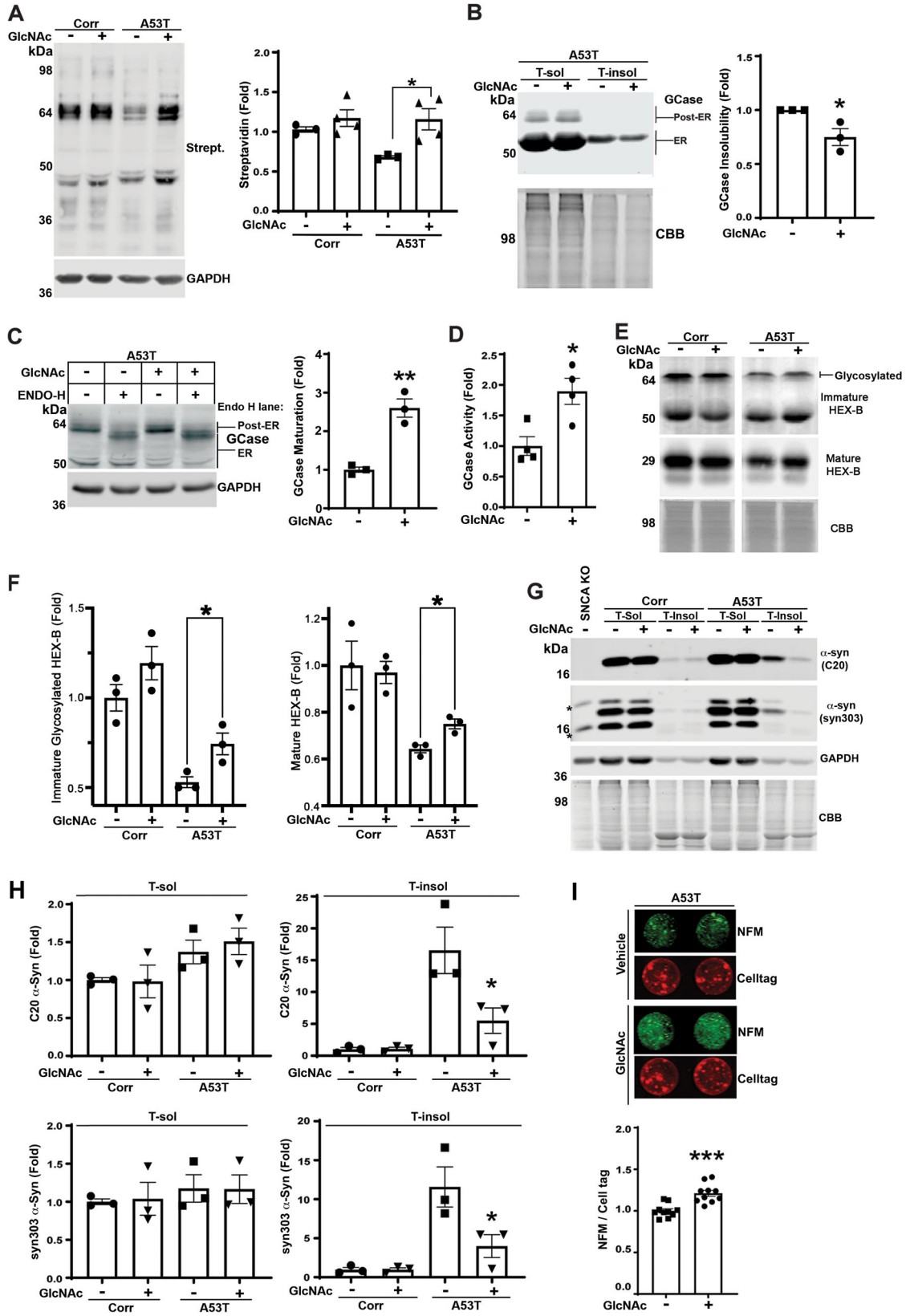

many studies have shown that ER-Golgi trafficking is perturbed in PD models[9,13,15–18,51], it is possible that failure of the UPR occurs through blocking ATF6 trafficking.

Our findings also indicate that enhancing the HBP may be a viable option to enhance lysosomal activity and clear protein aggregates in multiple neurodegenerative diseases. Seminal work on the HBP

pathway and proteostasis in *C. elegans* showed that HBP stimulation reduced protein aggregates in vivo[34]. Consistent with our findings, this study also showed that the IRE-1 branch of the UPR was required for GFAT-1 mediated rescue, given that IRE-1/XBP1s regulates the transcription of GFAT-1[33]. Furthermore, many downstream targets of the UPR including GRP78 (hsp-4 in *C. elegans*) were not upregulated by

**Fig. 6 | GlcNAc supplementation rescues N-glycosylation and downstream pathogenic phenotypes in PD iPSn. A** Western blot of N-glycosylated proteins using biotinylated Con-A, in iPSn (Corr and A53T iPSn at day 122) treated with 10 mM GlcNAc for 7 days. (−) indicates Vehicle control, which is PBS. GAPDH is a loading control. Quantification is shown on the right (n = 3 for A53T Vehicle control and n = 4 for all other groups). **B** Western blot analysis of GCase from Triton X-100 soluble (T-sol) and insoluble (T-insol) fractions in Corr and A53T iPSn in the presence or absence of GlcNAc, using coomassie brilliant blue (CBB) as a loading control. Quantification is shown on the right (n = 3). **C** GCase maturation was measured by western blot using endoglycosidase H (endo H) digestion of lysates from Veh and GlcNAc supplemented A53T iPSn. GAPDH is a loading control. Quantification is shown on the right (n = 3). **D** Analysis of lysosomal GCase activity in living A53T iPSn after supplementation with GlcNAc. Fluorescent substrate degradation was evaluated in a microplate reader for 3 h. Activity in acidic compartments was determined by measuring the response to a lysosomal inhibitor bafilomycin A1 n = 4. **E, F** Western blot analysis of Hex B in Corr and A53T iPSn at day

122 in the presence or absence of GlcNAc, using coomassie brilliant blue (CBB) as loading control. Quantifications of mature and immature Hex B are shown in (**F**) (n = 3). Note that the Immature Hex B bands are shown at a higher intensity compared to the mature Hex B bands, to avoid saturation of the mature forms. **G, H** Western blot analysis of α-syn in Corr and A53T iPSn from Triton X-100 soluble (T-sol) and insoluble (T-insol) fractions using C20 or syn303 antibodies, in the presence or absence of GlcNAc. Coomassie brilliant blue (CBB) was used as loading control. α-Syn knock-out (*SNCA* KO) iPSn was used as a negative control. * indicates non-α-syn bands. Quantification is shown in (**H**) (n = 3). **I** Neuron viability was assessed in iPSn A53T at day 122 supplemented with GlcNAc, by quantification of neurofilament (NFM) content and normalized to cell volume using cell-tag. Quantification is shown below (n = 10). Scatter plots represent individual culture well replicates. For all quantifications, values are the mean ± SEM, *p < 0.05, **p < 0.01, ***p < 0.001. Student's two-sided *t*-test was used for (**A, B, C, D, E, F, I**) and ANOVA with Tukey's post hoc test was used for panel **H**. Source data are provided as a Source Data file.

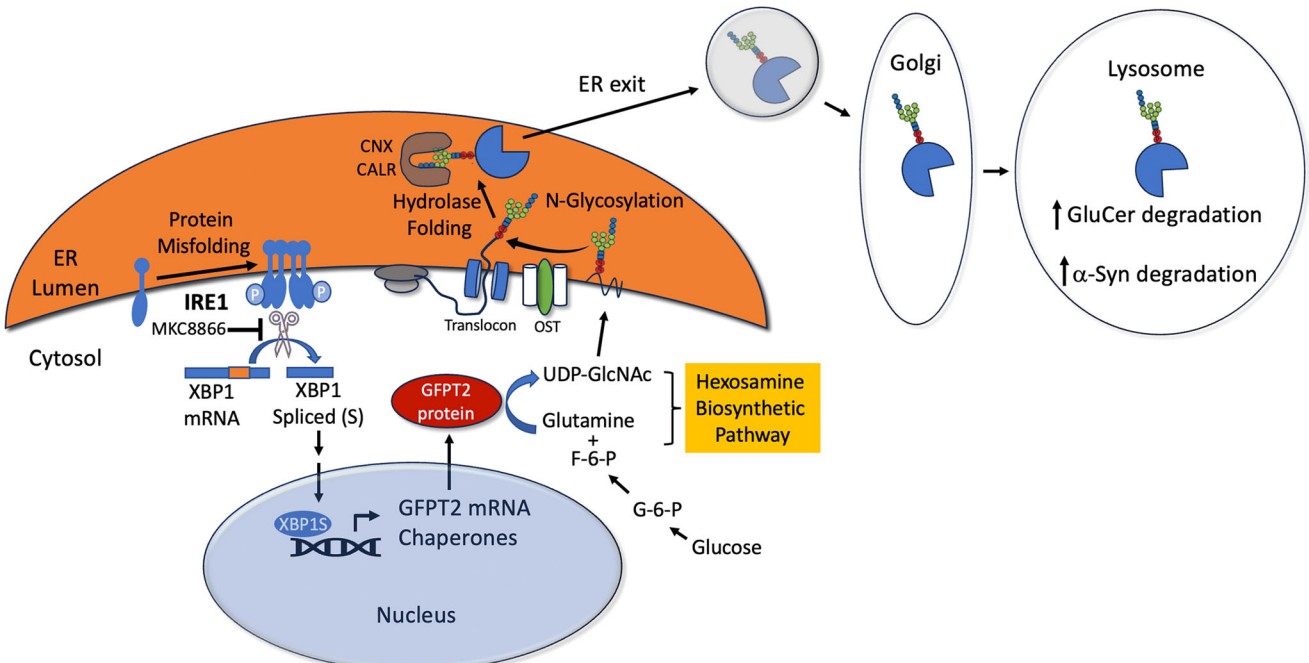

**Fig. 7 | The role of the Hexosamine Biosynthetic Pathway in integrating glucose flux and lysosomal function.** GFPT2 expression is triggered by the IRE1-XBP1s pathway under periods of protein misfolding stress in human midbrain dopamine neurons. MKC8866 is an inhibitor of IRE1 preventing both XBP1s and GFPT2 expression. The Hexosamine biosynthetic pathway utilizes cytosolic glucose and glutamine to generate N-glycan precursors that are used to glycosylate proteins in

the Endoplasmic Reticulum (ER) through the OST complex. Lectin chaperones including calnexin (CNX) and calreticulin (CALR) bind to N-glycans and promote proper folding of lysosomal hydrolases including GCase. Once properly folded, hydrolases can exit the ER and are transported through the Golgi then into lysosomes where they degrade substrates including glucosylceramide (GluCer in the case of GCase) and α-syn.

increased GFAT-1 activity, consistent with our findings (Figs. S5A, B, S6A). These data suggest that HBP-mediated rescue of lysosomal function and α-synuclein aggregation occurs through enhancing N-glycosylation, as opposed to indirect effects through enhancing ER chaperone expression.

Taken together, these findings highlight the fundamental role of the HBP in maintaining proteostasis in human iPSC-derived neurons and indicate that HBP failure plays an important role in neurodegeneration. We showed that global N-glycosylation reduction occurs in DLB brain using Con-A reactivity, while others have shown more complex changes in N-glycan patterns using different techniques in different neurodegenerative diseases[54,55]. This indicates that N-glycosylation changes may play a role in the pathogenesis of multiple diseases. For synucleinopathies, future work aimed at increasing the enzymatic activity of GFPT2, or enhancing flux downstream in the pathway with GlcNAc or potent derivatives, may provide a new class of

lysosomal enhancers to combat protein aggregation. Since altered glucose metabolism and protein aggregation are features of many neurodegenerative diseases, HBP enhancers may provide benefits in several neurodegenerative disorders. They may also be useful in slowing physiological aging by promoting proteostasis.

## Methods

All research complied with regulatory committees for research safety (Safety Protocol ID: BIO20230025). Procedures related to animal studies were approved under the Northwestern University IACUC protocol number IS00021691.

### Experimental model and sample details

**H4, Human neuroglioma cell culture.** Human H4 neuroglioma cells were maintained in Optimem media containing 5% fetal bovine serum (FBS), 200 μg/ml Geneticin and Hygromycin, and 1% penicillin/

streptomycin as described previously[9]. These cells express α-syn under the control of a tetracycline-responsive promoter, which can be turned off by the addition of 1 μg/ml doxycycline (DOX) in the media.

**SH-SY5Y cell culture.** Vector and α-syn expressing SH-SY5Y cells were generated as described previously[14]. These cells were cultured in DMEM supplemented with 10% FBS, 1% penicillin/streptomycin, and 200 μM G418. Differentiation was induced with all-trans-retinoic acid (10 μM) for 5 days.

**iPS cell culture and neuronal differentiation.** iPS cell culture and differentiation into midbrain dopaminergic neurons was done as previously published protocol[13]. Briefly, Human iPSCs were cultured on matrigel-coated 6 well plates in mTeSR1 media. Human iPS lines harboring the *SNCA* pathogenic variant G209A (A53T α-syn) and matching isogenic corrected lines (WIBR-iPS-SNCA[A53T] and WIBR-iPS-SNCA[A53T-Corr] as described in ref. 38) were generously gifted by Dr. R. Jaenisch (Whitehead Institute of MIT) and extensively characterized previously[38]. iPSC lines having *SNCA* gene triplication were previously described and extensively characterized[13,20]. iPSC differentiation into midbrain DA neurons was achieved using a mixture of growth factors for 40 days[56]. Neurons were maintained in neurobasal media (Thermofisher Scientific, #21103-049) containing NeuroCult SM1 supplement (Stem Cell Technologies, Inc. #05711) and 1% penicillin/streptomycin until used for experiments. Each batch of neurons was subjected to stringent quality control and were analyzed for maturation using the location of α-syn into synapse, by colocalization with synapsin by immunohistochemistry. The ratio of βiii-Tubulin/GAPDH was used to assess the efficiency of differentiation between batches.

**Control and transgenic α-synuclein mouse lines.** Synucleinopathy mouse model that expresses human A53T α-syn driven by the prion promoter (PrP) were previously described (line M83, B6; C3-Tg(Prnp-SNCA*A53T)83Vle/J, Strain #:004479)[39]. Brain samples were obtained from 10–14 month old symptomatic mice as described in the Figure legends from pathological regions cerebellum. The procedures were done as approved under the Northwestern University IACUC protocol number IS00021691. Control mice used here are of the same C3H/B6 background generated from crosses of A53T ± hemizygous mice, and housed under identical conditions as transgenic mice including the same holding room. Mice Housing, breeding, care, and use was as per the guidelines of Northwestern University's Institutional Animal Care and Use Committee and the US National Institutes of Health Guide for the Care and Use of Laboratory Animals. We used samples from both male and female mice, although our studies used $n = 3$ mice for each group and therefore we could not analyze statistical differences between males and females in this study. Our goal for this study was to define differences in GFPT2 levels based on the genotype alone. No sex differences in α-synuclein pathology or other phenotypes have been previously noted in this line.

**Human brain material.** De-identified samples from pathologically and clinically confirmed healthy controls and DLB human post-mortem frontal cortical tissues were obtained from the Northwestern University Alzheimer's disease pathology core (CNADC) (under the P30 grant P30AG013854), and The Brain Bank for Neurodegenerative Disorders at Mayo Clinic (under grants from Rainwater Charitable Foundation, Mangurian Foundation, State of Florida Alzheimer's Disease Initiative, and NIH grants P30 AG062677 and P01 AG003949). Samples were matched by age, sex, and post-mortem interval. Clinical data is available as Supplementary Data 1. We did not find any differences related to the sex/gender.

## Method details

**Materials.** Please refer to Table S1 for a detailed list of the reagents, source, and catalog numbers of items used in this study.

**Biochemistry and molecular biology. Removal of N-linked oligosaccharides from glycoproteins by PNGase F**

PNGase F was purchased from New England Biolabs (NEB-P0704s). Triton extracted cell lysates were subjected to PNGase digestion following the manufacturer's protocol. Briefly, 20 μg of lysate was mixed with 1 μl of Glycoprotein Denaturing Buffer (10×) and milliQ water to make a 10 μl total reaction volume. The proteins were denatured by boiling the samples at 100 °C for 10 minutes (min). Samples were then chilled on ice for 10 seconds (s), followed by addition of 2 μl GlycoBuffer 2 (10×), 2 μl 10% NP-40, and 6 μl H₂O. PNGase F (1 μl) was added and mixed gently. The reaction mix was incubated at 37 °C for 1 hour (h) followed by western analysis.

**Metabolic labeling for measurement of N-glycan incorporation into N-glycosylated proteins.**

Neurons were labeled with Ac4ManNAz (Sigma-900917-100 μM). This was followed by harvesting of neurons at 18, 36, 54, and 72 h. The cells were then extracted in Triton lysis buffer and protein assay was performed using DC™ Protein Assay Kit (Bio-Rad). The lysates were subjected to Biotin-phosphine reaction. In a reaction volume of 50 μl cell lysate, biotin phosphine (Cayman chemical-13581-50 μM) was added and incubated at room temperature overnight in a thermomixer. The reaction mix was subjected to western blot analysis using streptavidin-IRDye 800 conjugated detection reagent to detect biotinylated proteins (N-glycosylated).

**Immunoprecipitation of N-glycosylated proteins.** To a 500 μl neuronal lysate (1 μg/μl), 20 μg/ml biotinylated Concanavalin (Con-A) was added, and the reaction mixture was incubated overnight at 4 °C rotating end over end. NeutrAvidin agarose beads (29204, ThermoFisher, Scientific-25 μl) were used to recover Con-A bound proteins at 4 °C for 1 h. The mixture was spun at 2500 × *g* for 2 min to collect the beads. The beads were washed with PBS 3 times followed by elution of N-glycosylated proteins in 2X SDS buffer by boiling the samples at 95 °C for 10 min.

**Assessment of calnexin activity by Concanavalin-A pulldown.** *SNCA*-3X and A53T iPSn were infected with GFP or GFPT2 at MOI3, dpi 15, harvested, and extracted in 0.3% CHAPS buffer containing 40 mM HEPES pH 7.4, 120 mM NaCl, 1 mM EDTA, 10% vol/vol glycerol, protease inhibitor cocktail (PIC) (Roche), phenylmethylsulfonyl fluoride (PMSF) (Sigma), sodium orthovanadate (Na3VO4) (Sigma) and sodium fluoride (NaF) (Sigma). For pulldown of total N-linked glycosylated proteins, 400 μg lysate was mixed with 20 μg/ml biotinylated Concanavalin A (CON-A) (Vector Laboratories) and the reaction mixture was incubated overnight at 4 °C under gentle rotation. To recover CON-A bound proteins, 25 μl neutrAvidin agarose beads (Thermo Fisher Scientific) were added to the reaction mix and samples were incubated at 4 °C for 1 h. The beads were collected by centrifugation at 2500 × *g* for 2 min, followed by three washes with lysis buffer. N-glycosylated proteins were eluted by boiling the samples at 95 °C for 10 min in 2X Laemmli sample buffer. Samples were analyzed by western blot for calnexin (CANX), and GCase. CANX intensity in the pulldown lane was normalized to GAPDH levels from the corresponding inputs.

**Measurement of protein synthesis with SUrface SEnsing of Translation (SUnSET) assay.**

iPSn were treated with puromycin (5 μg/mL) for 1 h and harvested in cold PBS by centrifugation at 400 × *g* for 5 min. The cell pellets were subjected to homogenization in RIPA buffer (10 mM Tris/Cl pH 7.5, 150 mM NaCl, 5 mM EDTA, 0.1% SDS, 1% Triton-100 and 1% deoxycholate) and subjected to centrifugation at 21,000 × *g* for 20 min at

4 °C. The RIPA extracted lysates were incubated in ice-water slurry on a rocker for 30 min and then freeze thawed three times followed by ultracentrifugation at $100,000 \times g$ for 30 min at 4 °C. The protein concentration was measured by DC™ Protein Assay Kit (Bio-Rad). Lysates were subjected to SDS-PAGE and western blot analysis using anti-puromycin antibody.

**Targeted metabolomics.** iPSn in a 6 well format were placed on ice and quickly washed with ice-cold 0.9% NaCl. The plates were then placed on dry ice, followed by addition of 1 ml/well of 80% methanol extraction mix, containing isotopically labeled internal standard, Glutamate 13C5-15N (Cambridge Isotope Laboratories). Cells were thoroughly scrapped and transferred to pre-chilled Eppendorf tubes. The tubes were vortexed for 10 min at 4 °C. The samples were spun down at $21,000 \times g$ for 10 min at 4 °C, and the supernatant was speedvac dried and shipped on dry ice to the Metabolite Profiling Core Facility at the Whitehead Institute of the Massachusetts Institute of Technology (https://biology.mit.edu/tile/metabolite-profiling-core-facility). The pellet was resolubilized in 2% SDS lysis buffer and subjected protein assay by DC™ Protein Assay Kit (#500, Bio-Rad). The data was normalized to total protein. Relative abundance of each metabolite was obtained by UPLC-MS analysis using a Dionex UltiMate 3000 ultra-high performance liquid chromatography (UPLC) system using hydrophilic interaction chromatography (HILIC) and coupled to a Thermo Scientific QExactive Orbitrap mass spectrometer. Metabolite measurements were calculated by dividing the Raw peak area of each metabolite with the peak area of the internal standard—an isotopically labeled amino acid Glutamate 13C5-15N included in the extraction mix, which was then normalized to protein. For quality control (QC) analysis, a "pool" of sample consisting of a mixture of several μL from each of the biological samples was run, which created a representative sample that was run multiple times to get a measure of technical reproducibility for each metabolite. A CV (standard deviation/average) was calculated for these technical replicates and metabolites with a CV < 0.25 were considered reliably detected. Also, a larger injection volume was run with 0.3-fold and 0.1-fold dilutions of the pooled sample, which indicated whether samples fall in the linear range of detection for each metabolite, or there was a detector saturation or nearing the lower limit of detection. Metabolites that had a linear correlation coefficient $R < 0.975$ for this dilution series were flagged.

**Measurement of mRNA.** Total RNA was extracted and isolated using the PureLink RNA Mini Kit (Thermofisher Scientific). cDNA was synthesized using the RevertAid First Strand cDNA Synthesis Kit (Thermofisher Scientific). Real-time PCR was performed on an Applied Biosystems 7500 Fast system using pre-designed Taqman-primer probe sets: DPAGT1 (Hs00609752_m1), GFPT2 (Hs01049570_m1), XBP1U (Hs02856596_m1), XBP1s (Hs03929085_g1), GRP78 (Hs99999174_m1), HSP90B1/GRP94 (Hs00427665_g1), GBA1 (Hs00164683_m1), HexB (Hs01077594_m1) and ACTB (Hs01060665_g1). The quantifications represent the fold change of target mRNA expression normalized to beta-actin (ACTB) levels by delta-Ct method. The values are mean and s.e.m of biological replicates ($n = 3-6$) with two technical replicates for each.

**Sequential protein extraction and western blotting analysis.** Protein extraction was done as previously described in detail[57]. Briefly, midbrain neurons or mouse brain tissue were harvested in cold PBS by centrifugation at $400 \times g$ for 5 min. Supernatant was discarded and the cell pellets were subjected to homogenization in 1% Triton X-100 buffer (1% Triton X-100, 20 mM HEPES pH 7.4, 150 mM NaCl, 10% glycerol, 1 mM EDTA, 1.5 mM MgCl2, 1 mM phenylmethanesulfonyl fluoride (PMSF), 50 mM NaF, 2 mM Na orthovanadate, and a protease inhibitor cocktail (Roche diagnostics, https://www.roche.com, # 11-836-170-001)) by homogenization. The Triton extracted lysates were

incubated in ice-water slurry on a rocker for 30 min and then freeze thawed three times followed by ultracentrifugation at $100,000 \times g$ for 30 min at 4 °C. The supernatant (Triton soluble) was collected, and Triton-insoluble pellets were further extracted in 2% SDS lysis buffer by boiling for 10 min, followed by sonication and then ultracentrifugation at $100,000 \times g$ for 30 min at 22 °C. The protein concentrations of the Triton and SDS fractions was measured by DC™ Protein Assay Kit (Bio-Rad). Extracted protein lysates were subjected to SDS-PAGE followed by transfer of proteins onto a PVDF membrane (0.45 um pore size; Millipore), and post-fixed in 0.4% paraformaldehyde. Membranes were blocked in a 1:1 mixture of 1X TBS and Odyssey blocking buffer (Li-Cor Biosciences) for 1 h at RT, followed by overnight incubation with primary antibodies at 4 °C diluted in a 1:1 mixture of 1X TBS-Tween and Odyssey blocking buffer. The following day, secondary antibodies were added for 1 h, and the membranes were scanned using a Li-Cor Biosciences infrared imaging system or Azure Biosciences Sapphire scanner. Protein band intensities was quantified by the ImageStudio software version 3.1. We utilize two-color secondaries to simultaneously detect anti-mouse or anti-rabbit primary antibodies on the same membrane. Antibody information is listed in Table S1.

Frozen cerebellum dissected from control and A53T mice were homogenized in 1% Triton X-100 buffer in 1:5 weight/volume ratio. The lysates were incubated in ice-water slurry for 30 min, subjected to two freeze/thaw cycles, and ultracentrifuged at $100,000 \times g$, 4 °C for 30 min. Supernatant (Triton-soluble fraction) was subjected to SDS-PAGE followed by western blot analysis in the same way as culture lysates.

**Analysis of human brain tissue.** Control and DLB human post-mortem frontal cortical tissues were obtained from the Northwestern University Alzheimers disease pathology core or Mayo Clinic Pathology Core (Jacksonville, FL) and sequentially extracted by a 5-step extraction protocol using high salt buffer, 1% Triton X-100, 1% Triton + 30% sucrose (Sigma) and 1% sarkosyl (Sigma). Briefly, brain tissues were homogenized in high-salt buffer (HSB) (50 mM Tris-HCl pH 7.4, 750 mM NaCl, 10 mM NaF, 5 mM EDTA) with protease and protein phosphatase inhibitors, followed by incubation on ice for 20 min and centrifugation at $100,000 \times g$ for 30 min at 4 °C. The pellets were then re-extracted with HSB, followed by sequential extractions with 1% Triton X-100-containing HSB and 1% Triton X-100-containing HSB with 30% sucrose. The pellets were then resuspended and homogenized in 1% sarkosyl-containing HSB, rotated at 4 °C overnight, and centrifuged at $100,000 \times g$ for 30 min. The resulting sarkosyl-insoluble pellets were washed once with PBS and resuspended in PBS by brief sonication. The 1% Triton fraction was subjected to western blot analysis for the detection of GFPT2 protein or N-glycosylated proteins using Con-A Biotin with Streptavidin-conjugated secondary reagents.

**Analysis of post-ER GCase:** This assay was performed as described in detail[46]. Briefly, 40 μg of lysate were subjected to Endoglycosidase H (Endo H) digestion as per the manufacturer's protocol (New England Biolabs, https://www.neb.com) at 37 °C for 2 h. Duplicate samples without enzyme were incubated under the same conditions serving as control (undigested samples). The digestion was stopped by the addition of 5 X Laemmeli sample buffer, digested and undigested samples were run on 10% SDS-PAGE gels for 3–4 h at 120 V. This was followed by western blot analysis using anti-GCase (#G4171, Sigma, 1:500) antibody. Fluorescent secondary antibodies were used for detection (IRDye800 conjugated anti-rabbit, Licor Biosciences). Endo H resistant bands migrating 62–64 kDa were measured as post-ER forms and maturation was calculated by quantifying the post-ER/ER ratios. Quantification was performed on Image Studio software (Licor). GAPDH was used as a loading control. The measurements were done in triplicate for 2-to-3 distinct primary culture batches. Standard error of the mean values were graphed, analyzed by student's $t$-test, and $p < 0.05$ were considered significant.

**Live-cell lysosomal GCase activity assay.** The procedure and analysis method for the activity assay has been previously described in detail[46]. Briefly, 1 day prior to the assay, cells were treated with 1 mg/ml cascade dextran blue (Life Technologies) in a 96-well plate for 24 h. This was followed by treatment of cells with DMSO or 200 nM bafilomycin A1 for 1 h at 37 °C.The cells were subjected to 1 h pulse chase with an artificial fluorescent GCase substrate, 100ug/ml 5-(pentafluorobenzoylamino) fluorescein di-ß-D-glucopyranoside (PFB-FDGluc; Life Technologies), at 37 °C. The fluorescence signal was measured every 30 min for the span of 3–4 h on a plate reader (Ex = 485 nm, Em = 530 nm, for the GCase substrates; Ex = 400 nm, Em = 430 nm for cascade dextran blue). For the analysis, the GCase fluorescence signal was normalized to either lysosomal mass by using cascade dextran blue signal or total cell volume by quantifying CellTag 700 staining signal.

**In vitro GCase activity assay.** As described previously[9,58], GCase activity was measured from whole-cell Triton soluble lysates, using 4-methylumbelliferyl β-D-glucopyranoside (4-MU-Gluc; Sigma-Aldrich). In brief, 2–5 µg of total protein lysate was mixed with 1% BSA, 1 mM 4-MU-Gluc in activity assay buffer (0.25% (v/v) Triton X-100 (Sigma-Aldrich, #T-8787), 0.25% (w/v) Taurocholic acid (Sigma-Aldrich, # T9034), 1 mM EDTA, in citrate/phosphate buffer, pH 5.4) and incubated for 30 min at 37 °C. The reaction was stopped by adding equal volume of 1 M glycine, pH 12.5. The fluorescence (ex = 355 nm, em = 460 nm) was detected in a Molecular Devices i3 microplate reader using fluoro plates (Nunc, #475515). To control for assay and substrate specificity, parallel samples were additionally incubated with 200 µM of GC inhibitor CBE. Relative fluorescent units (RFUs) from CBE-treated lysates were subtracted from non-CBE-treated lysates. Obtained GCase activity was normalized to protein amount and is expressed as fold change.

**Glucose measurement.** Glucose levels were measured in 0.1% Triton soluble lysates by Glucose Assay Kit-WST (Water-soluble Tetrazolium salt (DOJINDO, Japan, Product code #G264), as per the manufacturers protocol. A53T iPSn (at day 90) infected with lentivirus expressing GFP (Control) and GFPT2 at MOI-3, 15 days after infection, were homogenized in 0.1% Triton X-100 buffer. The triton extracted lysates were incubated in ice-water slurry on a rocker for 30 min and then freeze thawed three times followed by ultracentrifugation at 100,000 × g for 30 min at 4 °C. The supernatant (Triton soluble) was collected, and protein concentrations was measured by DC™ Protein Assay Kit (Bio-Rad). Sample (20 µg) or glucose standard solutions, 50 µL each, were added to individual wells of 96-well microplate followed by the addition of working solution (50 µL). The assay plate was incubated at 37 °C for 30 min. Absorbance at 450 nm was measured with a microplate reader and the concentrations of glucose in the samples were calculated from the calibration curve.

**Imaging analysis**
**Neurofilament toxicity assay.** iPSC-neurons 96-well format were fixed in 4% paraformaldehyde in PBS for 20 min, followed by incubation in 0.3% Triton X-100 (PBS) for 20 min. The cells were then blocked with Odyssey blocking buffer (Li-Cor) for 1 h. Anti-neurofilament antibody (1:1000, mouse IgG 2H3, Developmental Studies Hybridoma Bank, University of Iowa, Iowa City, IA) was incubated overnight in blocking buffer at 4 °C, followed by washing in PBS with 0.1% Tween for 20 min. IRdye 800-conjugated anti-mouse IgG antibodies (1:1000 dilution, Li-Cor) was incubated in blocking buffer for 1 h, and CellTag™ 700 (Li-Cor) was also added for normalization. The cells were washed four times in PBS-0.1% Tween and scanned on an Odyssey infrared imaging system (Li-Cor). Neurofilament intensity was determined by Image Studio software (version 2.1 Li-Cor) and normalized to cell volume.

**Lentivirus generation and transduction**
**Generation of the GFPT2 plasmid and transduction in iPSn.** pENTR223-GFPT2 (ccsbBroadEn_07515) containing cDNA plasmid obtained from DNASU plasmid repository (http://dnasu.org/DNASU/GetCloneDetail.do?cloneid=514451#sequence). A stop codon was inserted by site-directed mutagenesis (SDM) using following mutagenesis primers (5′-CAAGTCTGTAACTGTGGAATGACCAACTTTCTTG-TACAAAGT-3′ and 5′-CAACTTTGTACAAGAAAGTTGGTCATTCCACA GTTACAGACTTG-3′). SDM was performed using the materials and procedures from the QuikChange XL Site-Directed Mutagenesis Kit (Agilent). The stop codon insertion was confirmed by sequencing. pENTR223-GFPT2 with stop codon was subcloned into pER4 for lentivirus production.

**Lentiviral overexpression constructs and preparation.** Lentiviral construct overexpressing XBP1s (VB900007-1013-https://en.vectorbuilder.com/vector/VB900007-1013mfa.html) was obtained from Vector Builder. Lentiviral construct overexpressing HA-DPAGT1 (https://gentaur.com/abm-lentivectors/dpagt1-lentiviral-vector-human-cmv-plenti-giii-cmv-c-term-ha-01011340106) was obtained from Applied Biological Materials Inc. (ABM). Lentivirus production was achieved by transfecting HEK293FT (Invitrogen, R70007) cells with the lentiviral plasmid expressing the target along with a packaging vector (psPAX2) and an envelope vector (VSV-G). The transfection was performed with X-tremeGENE transfection reagent (Roche #6366236001) as described previously[14]. Lentiviral particles were collected from the media and concentrated using Lenti-X concentrator (Clontech, #631232). The number of viral particles was determined by HIV1-p24 antigen ELISA kit (ZeptoMetrix, #801111) as per the manufacturer's protocol. The viral particle number and the number of plated cells was used to calculate the multiplicity of infection (MOI). For GFPT2 overexpression in A53T iPSn, we infected at MOI-3 and harvested 2 weeks post-infection. A53T neurons were infected with XBP1s lentivirus at MOI-1 and harvested 16 days post-infection. iPSn infected with lentivirus expressing GFP was used as control.

**Pharmacology**
**Tunicamycin treatment for inhibition of N-glycosylation.** Neurons were treated with or without 5 ug/ml tunicamycin (EMD Millipore/Calbiochem) for 24 h in DMSO and compared to equal amount of DMSO vehicle alone. Cells were extracted in 1% Triton X-100 buffer and analyzed by western blot for Hex B, Con-A, or GFPT2 as described in "Sequential protein extraction and western blotting analysis". For mRNA measurement, total RNA was extracted and isolated as described in "Measurement of mRNA." cDNA was synthesized and subjected to real-time PCR using Taqman-primer probe sets for GFPT2, XBP1U, and GRP78.

**N-Acetylglucosamine (GlcNAc) treatment.** iPSC neurons were supplemented with or without GlcNAc (A3286-sigma) at 10 mM for 7 days in PBS. Media was changed every other day for the duration of the experiment. The Vehicle control is PBS for each experiment, which is indicated as (-) symbol in the figures.

**Statistics and reproducibility**
In each quantification, a single plot point indicates a separate biological replicate (individual culture well, distinct mouse brain sample, or distinct human brain sample). The quantitative data are taken from at least $n = 3$ biological replicates and two distinct iPSC passages/differentiation batches for each assay to determine reproducibility between culture batches. The value of n and what n represents is indicated in each figure legend. Analyzed data was plotted and tested for statistical significance using the GraphPad Prism software. Statistical significance between two samples was determined using an unpaired t-test that is two-sided. For more than two conditions, significance was determined

using a one-way ANOVA with Tukey's multiple comparison test. For Fig. 1D, ANOVA with Šídák's multiple comparisons test was used. Sample sizes were chosen based preliminary studies as well as on the expected variability between culture sets and assays based on our previous publications[10,14,20,59]. Samples used to measure metabolites in Fig. 2 were blinded to the analyzer. For all other experiments, the investigators were not blinded to allocation during experiments and outcome assessment. It was not possible to blind samples in some cases, where cell lines or treatment conditions could be easily identified based on their respective unique, inherent features alone. No statistical method was used to predetermine sample size. iPSC midbrain cultures were plated in random areas of the culture well and, when applicable, randomly assigned a treatment or control group. Our study was designed to compare *SNCA* mutant lines with their respective isogenic corrected lines, using at least $n = 3$ biological replicates for each assay. A $p$ value of <0.05 was considered significant (*$p < 0.05$, **$p < 0.01$, ***$p < 0.001$, ****$p < 0.0001$). For each quantification, the type of error bar used and statistical test is specified in the figure legends. Samples were excluded only when technical outliers or odd culture sets were easily identifiable that include mis-loaded gels or transfer issues, cultures that did not pass our quality control measures as described under "iPS cell culture and neuronal differentiation", and numerical outliers were identified by GraphPad Prism using Grubb's test.

### Reporting summary

Further information on research design is available in the Nature Portfolio Reporting Summary linked to this article.

## Data availability

Source Data are provided as a Source Data file. The chromatogram files/Spectral data can be obtained from the MIT metabolomics core, Job1351 (wibr-metabolomics@wi.mit.edu). Source data are provided with this paper.

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

## Acknowledgements

This work was supported by the National Institute of Neurological Disorders and Stroke grant number R01NS092823 (J.R.M.) and the Michael J. Fox Foundation number MJFF-021532 (W.Y.W. and J.R.M.). We thank the Northwestern Center for Cognitive Neurology and Alzheimer's Disease Center (CNADC) (P30 grant P30AG013854) and The Brain Bank for Neurodegenerative Disorders at Mayo Clinic, directed by Dr Dennis W. Dickson and curated by Dr Michael DeTure with support from Mayo Clinic, Rainwater Charitable Foundation, Mangurian Foundation, State of Florida Alzheimer's Disease Initiative, and NIH grants P30 AG062677 and P01 AG003949 for providing human brain samples. We thank Caroline Lewis and the Metabolite Profiling Core Facility at the Whitehead Institute for running metabolomics samples.

## Author contributions

Conceptualization, W.Y.W., F.Z., J.R.M; Methodology, Validation, Formal Analysis, and Investigation, W.Y.W. F.Z., N.R.B., J.R.M., W.Y.W. performed and analyzed all of the experiments, except for: F.Z. performed experiments in Figure S5 and N.R.B. performed live-cell GCase activity assay for Fig. 6. Writing—Original Draft, W.Y.W. and J.R.M.; Writing—Review & Editing, W.Y.W. F.Z., N.R.B., J.R.M., Visualization, W.Y.W., and J.R.M., Supervision, J.R.M., Project Administration, J.R.M., Funding Acquisition, J.R.M.

## Competing interests

W.Y.W. and J.R.M. are co-inventors on a patent related to this work "Methods to synergistically enhance multiple cellular proteostasis pathways to treat neurodegeneration and storage diseases". The patent covers the use of N-glycosylation enhancers to treat or prevent a disorder associated with α-synuclein accumulation. United States Provisional Patent Application Ser. No. 63/264,224 (status pending) filed by Northwestern University. J.R.M. and W.Y.W. are among the inventors on this patent. N.R.B. and F.Z. declare no competing interest.
