## [Peer Review File · Nature Communications]

The hexosamine biosynthetic pathway rescues lysosomal dysfunction in Parkinson's disease patient iPSC-derived midbrain neuronsREVIEWER COMMENTS

Reviewer #1 (Remarks to the Author):

Wani et al investigated the role of the hexosamine biosynthetic pathway (HBP) in Parkinson's disease (PD). They show that N-glycosylation and glycan synthesis are impaired in A53T iPSn and transgenic mice. This is accompanied by a reduction in the level of GFPT2, a critical enzyme in the HBP pathway, both in A53T and SNCA triplication iPSn as well as in DLB brain. Finally, the authors show that both genetic enhancement and pharmacological intervention targeting the HBP pathway, promote N-glycosylation and the proper folding of lysosomal enzymes, such as hex-b and glucocerebrosidase (GCase).

This study expands on the seminal findings from Denzel et al that showed that increased synthesis of N-glycan precursors in the HBP improves proteostasis in the ER and extends lifespan in *C. elegans*. Overall, the observations of this study are interesting as this pathway could potentially play a role in the human aging process as well as neurodegenerative diseases.

The authors show that boosting the HBP pathway promotes the proper targeting and function of lysosomal enzymes such as hex-b and GCase. The main concern is that, in contrast to HexB, GCase is a poor substrate of GlcNAc-1-phosphotransferase as it is transported to the lysosomes in a man-6-P-independent pathway. This raises the question how the HBP pathway may impact GCase targeting. This may be an indirect effect that should be further explored. The HBP pathway improves the ER proteostasis machinery, which could indirectly affect lysosomal function. Another hypothesis could be that restoring the ER proteostasis by targeting the HBP promotes α -synuclein clearance that, in turn, facilitates proper lysosomal hydrolase folding, targeting, and function. Furthermore, whether/how this pathway is specific to PD and α -synuclein remains unclear. The mechanistic link between impaired glucose metabolism and HBP, lysosomal hydrolases, and α synuclein pathology requires additional experiments.

Finally, the mechanism linking the unfolded protein response and the HBP remains unclear and should be further examined.

Additional comments:

1. The authors state that each batch of neurons was subjected to stringent quality control. Representative images should be provided.
2. In Figure 1A the authors show that tunicamycin treatment has no effect on the N-glycosylation levels of A53T iPSn neurons. However, in Fig 1D Tunicamycin treatment leads to a decrease of the glycosylated form of the hydrolase HexB. Is this effect specific to hydrolase glycosylation? The authors should comment on this.
3. The authors hypothesise that N-glycosylation downstream of DPAGT1 is not perturbed. In addition to mRNA levels, a western blot showing DPAGT1 protein levels would be valuable.
4. Figure 1D: the authors should also evaluate GCase in this assay.
5. To strengthen the link between UPR and HBP, the authors should assess GFPT2 levels after XBP1s overexpression. Similarly, they should assess metabolite levels upon XBP1 overexpression (as in Fig 2B -

F).

6. The authors show that triggering the UPR through tunicamycin stimulates GFPT2 expression in controls, but not in PD neurons. In general, PD neurons are not responsive to tunicamycin. To strengthen these findings, an alternative ER stress inducer should be employed.

7. The authors should assess whether the genetic and/or pharmacological manipulation of the HBP pathway restores the metabolite changes observed in iPScn.

Concerning data analysis and statistics:

8. The authors should indicate the n in each figure legend.

9. Statistic is not clearly described in each figure legend.

10. It is unclear whether all the experiments were performed at d90.

11. Figure 3A: are data normalized over the untreated control?

Minor issues:

12. Please revise all the figure sequence cited in the text from 3D.

Reviewer #2 (Remarks to the Author):

In this manuscript, the authors describe a relationship between impaired hexosamine biosynthetic pathway (HBP) activity and lysosomal dysfunction in the context of Parkinson's disease. They demonstrate that deficiencies in UPR-dependent regulation of HBP genes, specifically GFPT2, leads to reduced N-linked glycosylation and subsequently impaired trafficking of lysosomal hydrolases in iPSc neurons expressing the PD-associated A53T α -synuclein mutant. Importantly, they show that overexpression of GFPT2 or enhancement of HBP function downstream of this enzyme afforded by exogenous administration of GlcNac can restore N-linked glycosylation and improve lysosomal hydrolase trafficking.

Overall, this is an interesting manuscript with clear conclusions derived from solid experiments. The link between deficiencies in UPR-dependent regulation of HBP and impaired lysosomal hydrolase trafficking is interesting and important. I only have a few comments about the work, most about the link between failed UPR signaling and HBP in neurons, that could be addressed in a revised submission, but, for me, this is a solid manuscript that would be well suited for publication in Nature Communications..

Comments.

1. I have some additional questions about the UPR-dependent regulation of HBP in neurons. In other cells, it is well established that the IRE1 arm of the UPR regulates HBP gene expression. It would assume that the same is the case for GFPT2 here, with the expression of the IRE1-regulated transcription factor XBP1s inducing this gene. This could be supported by showing that inhibition of IRE1 blocks Tm-induced increases in GFPT2. This would allow the authors to be a bit more explicit with respect to the link

between the UPR and HBP gene expression in this manuscript.

2. Along the same lines, does overexpression of XBP1s similarly improve N-linked glycosylation and lysosomal hydrolase trafficking in this model? They show nicely that GFPT2 expression improves these phenotypes, but this experiment could better highlight the link between failed UPR signaling and lysosomal dysfunction.

3. If I understand correctly, expression of A53T significantly reduces N-linked glycoproteins to levels similar to that observed in Tm-treated cells, yet protein synthesis rates are the same. This would suggest, as the authors indicate, that there would be an accumulation of non-glycosylated proteins in the ER, yet the UPR is not activated. I was curious if this reflects reduced expression of UPR sensors (e.g., IRE1, ATF6, PERK), or some other mechanism (e.g., impaired XBP1s nuclear import). This could be looked at by monitoring expression of these genes in RNA samples used in this paper. This doesn't have to be completely resolved for this paper, but it could be useful to address this in the manuscript.

4. Along the same lines as the above, I'm curious to know if Tm-induced UPR activation can be restored in A53T-expressing neurons expressing GFPT2 or treated with GlcNac.

5. Lastly, in a couple figures, there should be some additional annotation in the quantifications to make it clear what they are reflecting. E.g., the bar graph in 5A should indicate wt and A53T neurons.

As indicated above, these are just a few suggestions that could strengthen the link between failed UPR and HBP in this experiment, but overall this is an interesting manuscript that directly links deficiencies in HBP with lysosomal dysfunction in A53T asyn overexpressing neurons.

Reviewer #3 (Remarks to the Author):

The authors describe a series of experiments that more deeply investigate the role of the hexoseamine biosynthetic pathway (HBP) in synuclein deposition. This is supplemental to the authors recent publications on the same topics. This is generally a well-written manuscript that incorporates multiple models to validate their hypotheses and findings from a cellular basis. However, there are many sections of the manuscript that could be improved to provide clarity to the reader and consistency across the entire project. The authors suggest in the manuscript that the metabolomic analysis performed provides substantially new information. While these data tie together previous work, the metabolomic data are very limited.

The title of the manuscript does not offer much regarding new knowledge in this field. Altered and dysregulated glucose metabolism has been shown previously in PD and the HBP is known to be nutrient-sensing. Several recent (and many older) studies have shown altered glucose metabolism, shown the HBP is affected by and also affects glucose, and stating these are altered in PD is not, overall, a new concept. However, the rescue experiment employed in this study and the use of multiple models provides a nice addition to this knowledge and should be highlighted here. Although with similar findings to the multiple previously published works by the same group, including Stojkowska, Cuddy, and colleagues (2021, 2022).

There are several important concerns and questions that need to be addressed in the current manuscript as pointed out below, including multiple grammatical and typographical errors (an annotated file can be provided upon request).

1. The cell lines used in this study are key to the results. It is important that the cell lines are accurately described. The text suggests these are “patient neurons” or “patient cells” or “patient cultures”, but these are, in fact, patient-derived cell lines or corrected isogenic cell lines. Thus, the A53T patient-derived iPSC and neurons are “patient-derived,” they are not “patient cultures nor “patient neurons”. Please revise the text to ensure this is state accurately. Please ensure these models are properly cited. Authors mention the donated cells from Dr. Jaenisch but actually cite other papers in the text. Please confirm the cells that were used in this study--patient-derived (from fibroblast) and corrected A53T or the site-directed mutagenesis hiPSC lines generated by Soldner et al 2011?

2. Has the A53T iPS cell line been sequenced to consider modifiers?

3. Page 7: What is meant by “wildtype” iPSC-midbrain neurons, and how is this defined? What is the source of these neurons? How are you defining “human midbrain cultures” and “human midbrain neurons” are you distinguishing between these descriptions in the text?

4. Regarding mouse tissues that were “previously collected.” Are these mouse tissues from a published study? The origin of the tissues and how they were obtained should be described in more detail. Were wildtype littermates or other mice used as controls? The absence of any IACUC approval or other information for “tissues from the freezer” is concerning. How many mice? Sex? Age? The methods are not sufficient.

5. The targeted metabolomics methods and analyses are very limited and need more detail. Data are provided for only 5 molecules. The analysis must have been for a panel larger than this? or not? Please provide all metabolomic data.

6. Methods: What does “subject” mean the Methods section?

7. Methods: As above, transgenic mouse methods are inadequate.

8. Methods: Methods for metabolomics, as above, are inadequate. What was this targeted panel or certain molecules only were chosen? What were the standards? Where are the data?

9. References are not in proper format.

Figures:

Figure 3B: The GAPDH blot (first blot, top to bottom) looks like it is from a different experiment than the other 3 blots.

Figure 4A: The GFP blot does not appear to be the same blot.

Figure 4D: Are these blots all from the same experiment?

Figure 5C: No Veh is indicated. What is the vehicle?

Figure S2A: Is GAPDH from the same blot?

Figure S3D: The blot for a-syn is rather curved and not like the other blots cut out here. Please explain.

Comments below are related to clarity of the text:

1. Page 2, bottom: Please explain, “express wildtype GBA1 mutations.” Do you mean to say ..”do not express pathogenic GBA1 variants.”?
2. Throughout the manuscript: Since the terminology “A53T” is historical, I suggest using approved human genome nomenclature for all variants – at least at the beginning and in the methods so that there is no confusion and all readers are included.
3. Throughout the manuscript: Replace “mutation” with “variant” or “pathogenic variant” depending on context. This is the currently accepted terminology.
4. Throughout the manuscript, please be consistent: (a) in terminology: a-syn or alpha-syn or a-syn. Choose one and use it. (b) with all abbreviations: hours, h; minutes, min, etc. (c) in all punctuation and spacing: i.e. 10mM or 10 mM; n=3 or n = 3; p<0.005 or p < 0.005, etc. (d) define all abbreviations at first use

Response to Reviewers:

We thank the reviewers for carefully evaluating our manuscript and for suggesting additional work to strengthen the paper. Below, we reply to each reviewer's concern in a point-by-point manner.

Reviewer #1 (Remarks to the Author):

Wani et al investigated the role of the hexosamine biosynthetic pathway (HBP) in Parkinson's disease (PD). They show that N-glycosylation and glycan synthesis are impaired in A53T iPSn and transgenic mice. This is accompanied by a reduction in the level of GFPT2, a critical enzyme in the HBP pathway, both in A53T and SNCA triplication iPSn as well as in DLB brain. Finally, the authors show that both genetic enhancement and pharmacological intervention targeting the HBP pathway, promote N-glycosylation and the proper folding of lysosomal enzymes, such as hex-b and glucocerebrosidase (GCCase).

This study expands on the seminal findings from Denzel et al that showed that increased synthesis of N-glycan precursors in the HBP improves proteostasis in the ER and extends lifespan in *C. elegans*. Overall, the observations of this study are interesting as this pathway could potentially play a role in the human aging process as well as neurodegenerative diseases.

The authors show that boosting the HBP pathway promotes the proper targeting and function of lysosomal enzymes such as hex-b and GCCase. The main concern is that, in contrast to HexB, GCCase is a poor substrate of GlcNAc-1-phosphotransferase as it is transported to the lysosomes in a man-6-P-independent pathway. This raises the question how the HBP pathway may impact GCCase targeting. This may be an indirect effect that should be further explored.

Response:

Although HexB is transported by Man-6-Phos receptor and GCCase is transported via LIMP-2, our data indicates that the HBP promotes lysosomal targeting of both enzymes by promoting proper folding in the ER, which occurs prior to the transport step referred to by the reviewer. Both HexB and GCCase rely on N-glycosylation for proper folding in the ER, and only properly folded hydrolases can undergo ER exit. Folding occurs through the interaction of N-glycosylated GCCase with lectin chaperones in the ER including Calnexin / Calreticulin (Tan et al, Chemistry & Biology, 21, 967–976, August 14, 2014). Our previous work showed that immature GCCase accumulates and misfolds in the ER of PD neurons (Stojkovska et al, Neuron, 2022), and we show here that this ER retention occurs from reduced N-glycosylation. Once glycosylation of GCCase is increased (through GFPT2 or GlcNAc), it improves the properly folded, enzymatically active conformation (as shown in Fig 5G, 6D) which can then be transported by into lysosomes by either M6PR or LIMP2.

The HBP pathway improves the ER proteostasis machinery, which could indirectly affect lysosomal function.

Response:

We agree that it is important to consider indirect pathways that may enhance lysosomal function. To address this, we quantified the levels of ER proteostasis machinery (mRNA and protein) after HBP was enhanced by GlcNAc or GFPT2 expression. GFPT2 expression alone did not elevate the mRNA levels of GRP78, GRP94, or XBP1s (new Supplemental Figure 5A, shown below for convenience). We found no change in the mRNA or protein levels, even though the levels of N-glycosylated proteins were increased (Figure 5C). This suggests that the effects on lysosomal hydrolases occur through increasing N-glycosylation as opposed to indirect effects mediated by ER chaperones alone.

We next determined if increasing the HBP influences the levels of machinery involved in chaperone-mediated autophagy (CMA), since this a main way that physiological a-synuclein clearance occurs (Cuervo et al, Science, 2004). Measurement of HSC-70, which is an established marker of CMA that indicates the basal level of CMA in the cell, showed no change with GFPT2 (Reviewer Figure 2). This indicates that GFPT2 overexpression does not increase a-synuclein clearance by upregulating CMA.

Finally, we measured the levels of ER Chaperones and UGGT1 by western blot, after GlcNAc supplementation in A53T iPSc (matching the treatment conditions of Figure 6 in the manuscript). We found no change in GRP78, GRP94, or UGGT1.

Another hypothesis could be that restoring the ER proteostasis by targeting the HBP promotes α -synuclein clearance that, in turn, facilitates proper lysosomal hydrolase folding, targeting, and function.

Response:

We do agree that once α -synuclein levels are reduced in cell, it will help to facilitate hydrolase trafficking, but this would occur at the cis-Golgi when ER-derived vesicles fuse to the Golgi. Our data accumulated over the past several years indicate that α -syn can disrupt trafficking by blocking an ER-Golgi SNARE called ykt6 at the cis-Golgi (Cuddy et al, Neuron, 2019; Mazzulli et al, PNAS 2016). We have also shown that reducing α -synuclein by other methods will facilitate proper hydrolase folding (eg, Mazzulli et al, PNAS 2016 & Mazzulli et al J. Neurosci 2016). However in the context of this study, α -synuclein reduction occurs downstream of HBP enhancement, since we know that N-glycosylation promotes proteostasis in the ER, prior to ER-Golgi transport. When GFPT2 is overexpressed in PD iPSc, the trigger that initiates the α -synuclein reduction is increased lysosomal activity (through an enhancement of N-glycosylation of hydrolases). Our new data in Supplemental Figure 6C (shown below for convenience) shows that reducing lysosomal activity with leupeptin treatment prevents the clearance of α -synuclein. This indicates that GFPT2 requires functional lysosomes to reduce α -synuclein.

Furthermore, whether/how this pathway is specific to PD and a-synuclein remains unclear.

Response:

To address this, we analyzed the levels of N-glycosylated proteins in different neurodegenerative diseases. We analyzed cortical post-mortem brain lysates from Alzheimer's disease (AD), progressive supranuclear palsy (PSP), brainstem Lewy body disease (BLBD), and Dementia with Lewy bodies (DLB) using con-A / western blot. Note that BLBD only shows a-syn pathology in the brainstem while the cortex is free of pathology. We find that only DLB samples show reduced N-glycosylated proteins. This data is included as Figure 1, panel C of the new draft (shown below for convenience). This indicates that reduced N-glycans are specific for synucleinopathies.

The mechanistic link between impaired glucose metabolism and HBP, lysosomal hydrolases, and alpha synuclein pathology requires additional experiments.

Response: We have done several additional experiments to further strengthen the mechanistic link between the HBP and lysosomal clearance of a-syn. First, as mentioned above, we find that reduced N-glycosylated proteins occur only in patient brain that exhibits Lewy (a-syn) pathology but not abeta or tau pathology (as seen in AD and PSP). This indicates that the hypo-N-glycosylation phenotype does not occur from general neurodegeneration, but is specific for synucleinopathies.

Furthermore, we have added new data to our rescue experiments showing that restoring the HBP can rescue lysosomes. We find that increasing the rate-limiting enzyme in the HBP (GFPT2) or GlcNAc, can rescue lysosomal function in PD neurons (Figure 5 &6). New data shows that GFPT2 overexpression also reduces intracellular glucose levels in PD neurons (Figure 5B), demonstrating that glucose accumulation occurs in part through dysfunction of the HBP. GFPT2 and GlcNAc both enhance hydrolase maturation and activity within lysosomal compartments (Figure 5D, E, G). New experiments show that GFPT2 has no effect on CMA machinery or ER chaperone levels, demonstrating that the syn reduction is a direct result of improving N-glycosylation of lysosomal hydrolases and downstream function (Reviewer Figures 1, 2 above). This is supported by data in figure S6B above where inhibiting lysosomes prevents a-syn clearance. We also demonstrated that DPAGT1 overexpression has no effect on lysosomal phenotype or a-synuclein levels (new Supplemental figure S4), further validating that the perturbations of the HBP occur upstream of DPAGT1.

Finally, the mechanism linking the unfolded protein response and the HBP remains unclear and should be further examined.

Response: To address this, we have done several additional experiments. First, we found that inhibiting IRE1 activity and reducing XBP1s prevents the upregulation of GFPT2 during periods of ER-stress (induced by tunicamycin). This is shown in new figure 3 E, F.

Next, we expressed a misfolded mutant form of GCase (L444P) that causes Gaucher disease in both control and PD iPScn. While control neurons responded to misfolded GCase by upregulating XBP1s and GFPT2, PD neurons showed no response (new figure 3G, H, shown below for convenience).

Finally, we include new Figure 4, which demonstrates that direct expression of the active XBP1 transcription factor (XBP1s) can rescue GFPT2, N-glycosylation, lysosomal enzyme activity and reduce a-syn levels. Together, these data strengthen the mechanistic connection between the UPR and the HBP in PD.

Additional comments:

1. The authors state that each batch of neurons was subjected to stringent quality control. Representative images should be provided.

Response: To address this, we now provide representative images below of immunostaining results and immunoblots that we use to routinely authenticate the material (Reviewer Figure 8). The midbrain protocol that we use has been standardized and we perform QC analysis at 50-to-60 days in culture, prior to doing in-depth studies. We quantify the percentage of cells that express midbrain DAergic markers, including tyrosine hydroxylase (TH), FOXA2, b-iii-Tubulin, and others not shown below such as LMX1a. Given that the differentiation protocol is now standard procedure in the lab, and our group has already published representative images of iPSC-derived midbrain cultures, we did not include these data in the main body of the manuscript. However, in addition to the images below, which are from the A53T iPScn and isogenic corrected lines used throughout the manuscript, the reviewer is referred to images in our previous publications including Zunke et al, Neuron, 2018 (Supplemental Figure 4), Cuddy et al, Neuron, 2019 (Supplemental Figure 1), and Stojkovska et al, Neuron, 2022 (Supplemental Figures S1 and S2).

Reviewer Figure 8. Quality control analysis of iPSC-derived midbrain cultures. **A)** Immunostaining analysis of A53T and isogenic corrected (Corr) iPSC-derived neurons at day 60 showing TH and β-iiiTubulin staining. Dapi is shown in blue in the merged image. **B)** Immunostaining analysis for FOXA2 (red) and nuclei (DAPI) are shown in blue. We typically yield ~80-90% neurons that express FOXA2 + TH + βiii Tubulin. **C)** Western blot analysis of A53T and Corr lines. a-Syn was measured to assure that a phenotype exists. **D)** SNCA 3X cultures were analyzed as in C. Values are the mean, +/- SEM. N=4 culture wells.

2. In Figure 1A the authors show that tunicamycin treatment has no effect on the N-glycosylation levels of A53T iPSn neurons. However, in Fig 1D Tunicamycin treatment leads to a decrease of the glycosylated form of the hydrolase HexB. Is this effect specific to hydrolase glycosylation? The authors should comment on this.

Response: We do not think that the tunicamycin effect is specific for hydrolase glycosylation. The difference between Figure 1A and (new) Figure 1E is likely related to the sensitivity of the detection reagents used in each experiment. Fig 1A uses Con-A, which will detect all N-glycosylated proteins. Since con-A detects global N-glycosylation changes, it is likely not as sensitive for detecting changes in some specific proteins, which is why we performed targeted analyses on Hex B and GCase. The differences between the two panels is also likely due to individual synthesis rates and ER exit rates, which vary between different proteins (Lodish et al, Nature, Vol 304, July 1983). Our new data in Figure 1F show that tunicamycin has no effect on GCase maturation in A53T iPSn (shown below as reviewer figure 9). However, it is important to note that the main conclusion for all three readouts (Con-A, Hex B, and GCase) is consistent, which is that A53T iPSn show a blunted response to tunicamycin compared to Corr cells.

3. The authors hypothesise that N-glycosylation downstream of DPAGT1 is not perturbed. In addition to mRNA levels, a western blot showing DPAGT1 protein levels would be valuable.

Response: We agree and have now included DPAGT1 western blot (new Supplemental Figure S2C). We find that DPAGT1 levels are actually increased, which could be due to a compensatory mechanism, or result from reduced clearance of DPAGT1 protein. We find similar results when an HA-tagged plasmid is expressed in A53T iPSn (new Supplemental Figure S4A). We also tried to rescue the glycosylation and lysosomal phenotype by DPAGT1 overexpression, however this strategy failed to rescue A53T iPSn (Supplemental Figure S4). Collectively, these data further support that reduced glycosylation occurs from GFPT2 depletion, consistent with our original conclusions.

4. Figure 1D: the authors should also evaluate GCCase in this assay.

Response: We agree and now have assessed GCCase in new Figure 1E (shown below for convenience).

5. To strengthen the link between UPR and HBP, the authors should assess GFPT2 levels after XBP1s overexpression. Similarly, they should assess metabolite levels upon XBP1 overexpression (as in Fig 2B -F).

Response: We agree and to address this, we have expressed XBP1s in A53T iPSn and measured GFPT2 by lenti-viral plasmids. In New Figure 4, we found that XBP1s elevated GFPT2 mRNA and protein by ~ 50%. Although we ran into complications when trying to do metabolomic analysis for this (MIT core stop accepting external samples, and were also not able to yield sufficient material to measure metabolite levels by HPLC-MS analysis after XBP1s transduction), we did measure N-Glycosylated proteins and found that they were elevated (New Figure 4C). Given that N-Glycosylation is the end product of the HBP and requires HBP metabolites, we think it is also likely that those metabolites change upon XBP1s expression. Importantly, XBP1s also increased GCCase activity and reduced α -synuclein levels (New Figure 4D, E).

6. The authors show that triggering the UPR through tunicamycin stimulates GFPT2 expression in controls, but not in PD neurons. In general, PD neurons are not responsive to tunicamycin. To strengthen these findings, an alternative ER stress inducer should be employed.

Response: To address this, we now have overexpressed a Gaucher disease mutant form of GCCase (L444P) which we previously showed induces ER stress in GD neurons and healthy control neurons (Stojkovska et al, Neuron, 2022). Here, we find that expression of L444P elicits a UPR response in corrected iPSn, but not in PD iPSn (SNCA 3x) (new Figure 3G, H). We measured ER stress by XBP1s mRNA and GFPT2 mRNA. Our previous work measured multiple mRNA transcripts when stressed by L444P GCCase including GRP78 and GRP94, with similar results (Stojkovska et al, Neuron, 2022).

7. The authors should assess whether the genetic and/or pharmacological manipulation of the HBP pathway restores the metabolite changes observed in iPSn.

Response: We agree this is an important readout and to address this, we focused on measuring glucose levels after GFPT2 overexpression. New data in figure 5B show that GFPT2 reduces intracellular glucose levels by about 10%. Considering that the HBP normally utilizes about 5% of the total glucose, we feel this is a biologically significant change. This is supported by the fact that N-glycosylated proteins and lysosomal function were also elevated by GFPT2. Although a full metabolic analysis would have been optimal, limitations in the amount of iPSn cultures and in the MIT core as mentioned above prevented us from successfully completing this within the time frame of the revision. Instead, we opted to measure glucose and N-glycans, which can be quantified from a cellular lysate and used for other analyses such as multiple western blots, and was a more efficient use of the material. Our measurements of the earliest metabolite of the HBP (glucose) as well as the latest (N-glycans), both indicate a rescue effect by GFPT2.

Concerning data analysis and statistics:

8. The authors should indicate the n in each figure legend.

Response: We apologize for omitting this and have now updated the figure legends with this information. Scatter plots represent individual culture well replicates, individual mouse or human brain samples, which is now written in each figure legend.

9. Statistic is not clearly described in each figure legend.

Response: We apologize for omitting this and have now updated the figure legends with this information including when T-test and ANOVA with post-hoc test was used.

10. It is unclear whether all the experiments were performed at d90.

Response: We have updated the figure legends to clarify this point. Most of the data was analyzed when cultures were at day 90, and we mentioned this point in the first part of the figure (panel A) but did not repeat it throughout the figure legend for brevity. However some of our rescue experiments were analyzed 1 or 2 weeks later in order to allow for rescue effect to take place.

11. Figure 3A: are data normalized over the untreated control?

Response: Yes, the values from the A53T iPSn were normalized to the untreated control values (1 fold for each of the replicates).

Minor issues

12. Please revise all the figure sequence cited in the text from 3D.

Response: We thank the reviewer for catching this error and have now corrected it.

Reviewer #2 (Remarks to the Author):

In this manuscript, the authors describe a relationship between impaired hexosamine biosynthetic pathway (HBP) activity and lysosomal dysfunction in the context of Parkinson's disease. They demonstrate that deficiencies in UPR-dependent regulation of HBP genes, specifically GFPT2, leads to reduced N-linked glycosylation and subsequently impaired trafficking of lysosomal hydrolases in iPSC neurons expressing the PD-associated A53T α -synuclein mutant. Importantly, they show that overexpression of GFPT2 or enhancement of HBP function downstream of this enzyme afforded by exogenous administration of GlcNac can restore N-linked glycosylation and improve lysosomal hydrolase trafficking.

Overall, this is an interesting manuscript with clear conclusions derived from solid experiments. The link between deficiencies in UPR-dependent regulation of HBP and impaired lysosomal hydrolase trafficking is interesting and important. I only have a few comments about the work, most about the link between failed UPR signaling and HBP in neurons, that could be addressed in a revised submission, but, for me, this is a solid manuscript that would be well suited for publication in Nature Communications..

Response: We thank the reviewer for reading the manuscript and encouraging comments.

Comments.

1. I have some additional questions about the UPR-dependent regulation of HBP in neurons. In other cells, it is well established that the IRE1 arm of the UPR regulates HBP gene expression. It would assume that the same is the case for GFPT2 here, with the expression of the IRE1-regulated transcription factor XBP1s inducing this gene. This could be supported by showing that inhibition of IRE1 blocks Tm-induced increases in GFPT2. This would allow the authors to be a bit more explicit with respect to the link between the UPR and HBP gene expression in this manuscript.

Response: We thank the reviewer for this suggestion. Using control iPSc lines, we treated cultures with the IRE1 inhibitor MKC8866, in the presence or absence of tunicamycin. Q-RT-PCR to measure mRNA showed that XBP1s is reduced as expected, but also that GFPT2 mRNA was reduced (New Figure 3 E, F; also shown above as Reviewer Fig 6). This supports the idea that GFPT2 transcription is controlled by the IRE1 arm of the UPR, and is consistent with previous studies on GFAT1 in other cell types (Wang et al, Cell, 2014).

2. Along the same lines, does overexpression of XBP1s similarly improve N-linked glycosylation and lysosomal hydrolase trafficking in this model? They show nicely that GFPT2 expression improves these phenotypes, but this experiment could better highlight the link between failed UPR signaling and lysosomal dysfunction.

Response: To address this, we overexpressed XBP1s in A53T iPSc and found that it could elevate GFPT2 mRNA and protein, as well as improve N-glycosylation (using a Con-A blot). XBP1s also increased cellular GCCase activity and reduced a-synuclein (shown in New Figure 4).

3. If I understand correctly, expression of A53T significantly reduces N-linked glycoproteins to levels similar to that observed in Tm-treated cells, yet protein synthesis rates are the same. This would suggest, as the authors indicate, that there would be an accumulation of non-glycosylated proteins in the ER, yet the UPR is not activated. I was curious if this reflects reduced expression of UPR sensors (e.g., IRE1, ATF6, PERK), or some other mechanism (e.g., impaired XBP1s nuclear import). This could be looked at by monitoring expression of these genes in RNA samples used in this paper. This doesn't have to be completely resolved for this paper, but it could be useful to address this in the manuscript.

Response: We have measured the mRNA levels of the three main UPR sensors and find that while there is no change in IRE1 or ATF6, PERK mRNA is slightly elevated (Reviewer Figure 10). These data suggest that the failure of the UPR in PD iPSc is not due to reduced expression of the three sensors. Our previous work showed that the downstream target of PERK, pEIF2a, is not elevated in PD iPSc (Stojkowska et al, Neuron, 2022). Therefore, more work is required to determine at which step a-syn disrupts the UPR pathway. Since it seems that XBP1s expression can rescue many of the phenotypes observed here (New Figure 4), we hypothesize that the perturbation occurs prior to this step. A study from another group showed that nuclear translocation of ATF6 is inhibited by a-syn (Credle et al, Neurobiology of Disease, 2015), and this may also play a role in PD patient neurons since ATF6N can upregulate XBP1 mRNA. Future work is aimed at determining how PD cultures sense and respond to misfolded proteins, by focusing on the 3 main UPR sensors.

4. Along the same lines as the above, I'm curious to know if Tm-induced UPR activation can be restored in A53T-expressing neurons expressing GFPT2 or treated with GlcNac.

Response: We have measured the mRNA levels of GRP78 and GRP94 in GFPT2-infected A53T neurons +/- tunicamycin. We see that these cultures can respond by upregulating the mRNAs of these to UPR targets (Reviewer Fig 11). This suggests that UPR could be rescued, however, more studies are required to determine if the response is comparable to a healthy control iPSc line (we had attempted this experiment but did not have enough material from isogenic control neurons to compare). Given that a-synuclein levels are reduced by GFPT2 however, and our data indicate that a-syn accumulation is required to perturb UPR signaling since we are using isogenic control lines (this study, and Stojkowska et al Neuron 2022), we feel that the UPR response would eventually return to normal post-GFPT2 transduction.

Reviewer Figure 11. UPR measured by GRP78 and GRP94 mRNA after GFPT2 overexpression in A53T iPSc. Cultures at day 105 were transduced with GFPT2 plasmids and analyzed 2 weeks post infection by Q-RT-PCR (n=3-6 culture wells). **p<0.01, Student's T-test.

5. Lastly, in a couple figures, there should be some additional annotation in the quantifications to make it clear what they are reflecting. E.g., the bar graph in 5A should indicate wt and A53T neurons.

Response: We thank the reviewer for catching this error. We have now corrected this figure (now it is Figure 6A) and have gone through the other figures to make sure they are all labeled correctly.

As indicated above, these are just a few suggestions that could strengthen the link between failed UPR and HBP in this experiment, but overall this is an interesting manuscript that directly links deficiencies in HBP with lysosomal dysfunction in A53T asyn overexpressing neurons.

Response: We thank the reviewer for the constructive comments that have helped to further strengthen our paper.

Reviewer #3 (Remarks to the Author):

The authors describe a series of experiments that more deeply investigate the role of the hexoseamine biosynthetic pathway (HBP) in synuclein deposition. This is supplemental to the authors recent publications on the same topics. This is generally a well-written manuscript that incorporates multiple models to validate their hypotheses and findings from a cellular basis. However, there are many sections of the manuscript that could be improved to provide clarity to the reader and consistency across the entire project. The authors suggest in the manuscript that the metabolomic analysis performed provides substantially new information. While these data tie together previous work, the metabolomic data are very limited.

The title of the manuscript does not offer much regarding new knowledge in this field. Altered and dysregulated glucose metabolism has been shown previously in PD and the HBP is known to be nutrient-sensing. Several recent (and many older) studies have shown altered glucose metabolism, shown the HBP is affected by and also affects glucose, and stating these are altered in PD is not, overall, a new concept.

Response: To our knowledge, there has not been a previously published study that links HBP dysfunction to synucleinopathies or lysosomal dysfunction. Specifically, the identification of GFPT2 deficiency across PD models and DLB brain, and that GFPT2 enhancement could be a new therapeutic target to enhance lysosomal function, has not been previously shown.

However, the rescue experiment employed in this study and the use of multiple models provides a nice addition to this knowledge and should be highlighted here. Although with similar findings to the multiple previously published works by the same group, including Stojkowska, Cuddy, and colleagues (2021, 2022).

Response: We thank the reviewer for this suggestion and have changed to title to reflect the novel findings and study in human midbrain neurons: "The hexosamine biosynthetic pathway rescues lysosomal dysfunction in Parkinson's disease patient-derived midbrain neurons."

There are several important concerns and questions that need to be addressed in the current manuscript as pointed out below, including multiple grammatical and typographical errors (an annotated file can be provided upon request).

1. The cell lines used in this study are key to the results. It is important that the cell lines are accurately described. The text suggests these are "patient neurons" or "patient cells" or "patient cultures", but these are, in fact, patient-derived cell lines or corrected isogenic cell lines. Thus, the A53T patient-derived iPSC and neurons are "patient-derived," they are not "patient cultures nor "patient neurons". Please revise the text to ensure this is state accurately. Please ensure these models are properly cited. Authors mention the donated

cells from Dr. Jaenisch but actually cite other papers in the text. Please confirm the cells that were used in this study--patient-derived (from fibroblast) and corrected A53T or the site-directed mutagenesis hiPSC lines generated by Soldner et al 2011?

Response: We apologize for this lack of clarity and now have modified the text to reflect the fact that these are patient-derived iPSC-neurons. We have defined this early in the text and abbreviated “iPSn” throughout. In the first sentence of the results, we write “To assess changes in N-glycosylation, we measured the total levels of protein N-glycans in PD midbrain cultures derived from iPSC lines (iPSn) that express the disease-causing A53T a-syn mutation³⁷.” We either refer to the lines specifically (A53T or SNCA-3x) or “PD iPSn” when referring to data that includes both of these lines. Note that Reference 37 is Soldner et al, 2011 Cell paper, although in other areas of the text we cite our own papers that show pathological phenotypes in these lines, as we have been working with these same lines for over 10 years. We confirm that these are derived from a patient that harbors the A53T mutation, and not the site-directed mutagenesis lines, and this was clarified in the methods section. Specifically, the lines we received are called “WIBR-iPS-SNCA^{A53T} “ and “WIBR-iPS-SNCA^{A53T-Corr} “ as described in Soldner et al., Cell 2011 DOI: 10.1016/j.cell.2011.06.019 developed in the laboratory of Rudolf Jaenisch at the Whitehead Institute for Biomedical Research.

2. Has the A53T iPS cell line been sequenced to consider modifiers?

Response: Although the SNCA genomic region of both lines WIBR-iPS-SNCA^{A53T} “ and “WIBR-iPS-SNCA^{A53T-Corr} were originally sequenced by the Jaenisch group to confirm gene correction, to our knowledge, other PD modifiers were not considered. Therefore to address this concern, we subjected the parental line (WIBR-iPS-SNCA^{A53T} , denoted as “A53T iPSn” in our manuscript) to whole genome sequencing and searched for the top PD-associated single nucleotide polymorphisms (SNPs) from the most recent PD GWAS meta-analysis from the iPDGC Locus Browser (<https://pdgenetics.shinyapps.io/GWASBrowser/>)

We included the data on the next page for the top 55 PD-associated SNPs, including Odds ratios and P-values from the meta-analysis, and if the SNP is present in the A53T iPSn line (1/1 homozygous for alternative allele, 0/1 heterozygous, 0/0 homozygous for the reference allele, ./ missing alleles)

Although the A53T line harbors additional SNPs that could influence phenotypes, please note that we are comparing this A53T iPSn line to its isogenic control line in all assays. Therefore, since the genetic backgrounds are the same, the phenotypes we observe must occur from the presence of the highly pathogenic and penetrant A53T variant, and not the other less-penetrant SNPs listed below. Additionally, some of these genes are not expressed in neurons and would not be relevant for this study. We do agree it is important to consider these additional SNPs for future studies, as they may be involved in modulating the phenotypes we observe in our models.

Whole genome sequencing analysis of A53T iPSn identifies PD-associated variants.

hg38 chr bp REF ALT	Effect allele.x	risk allele	SNP	Nearest Gene	Beta all studies	OR all studies	SE all studies	P all studies	Genotype (A53T iPSn)
1 161499264 G C	C	C	rs6658353	FCGR2A	0.065	1.067	0.0094	6.1E-12	1/1
1 205754444 C T	T	T	rs823118	NUCKS1	0.1066	1.112	0.0094	1.11E-29	1/1
1 226728377 T C	T	T	rs4653767	ITPKB	0.0833	1.087	0.0104	1.38E-15	0/0
1 232528865 C T	T	T	rs10797576	SIPA1L2	0.1114	1.118	0.0133	6.84E-17	0/1
10 15515407 C T	T	T	rs896435	ITGA8	0.0735	1.076	0.0101	3.41E-13	0/1
11 10537230 A G	A	A	rs7938782	RNF141	0.087	1.091	0.0145	2.12E-09	0/1
11 133917106 G T	T	T	rs3802920	IGSF9B	0.1073	1.113	0.0117	6.26E-20	1/1
11 83776234 C A	A	C	rs12283611	DLG2	-0.0645	0.938	0.0102	2.61E-10	0/0
12 122842051 G T	T	T	rs10847864	HIP1R	0.1478	1.159	0.0115	1.47E-37	0/1
12 132487182 G A	A	A	rs11610045	FBRSL1	0.0601	1.062	0.0094	1.77E-10	0/1
14 37520065 T C	T	C	rs12147950	MIPOL1	-0.0529	0.948	0.0096	3.54E-08	0/1
14 87997920 G T	T	T	rs979812	GALC	0.061	1.063	0.0093	6.19E-11	0/1
15 61705186 T C	T	C	rs2251086	VPS13C	-0.1186	0.888	0.0137	6.08E-18	1/1
16 19266171 A T	A	A	rs6497339	SYT17	0.063	1.065	0.0095	2.76E-11	0/0
16 28933075 C G	C	G	rs2904880	CD19	-0.065	0.937	0.0106	7.87E-10	0/1
16 30966478 G A	A	A	rs11150601	SETD1A	0.0907	1.095	0.0099	5.12E-20	0/1
16 50702745 A G	A	A	rs6500328	NOD2	0.0586	1.060	0.0097	1.82E-09	0/0
17 44219699 A C	A	A	rs2269906	UBTF	0.0631	1.065	0.0102	6.24E-10	0/1
17 61840005 C T	T	T	rs61169879	BRIP1	0.082	1.085	0.0134	9.28E-10	0/1
17 78429399 A T	A	A	rs666463	DNAH17	0.076	1.079	0.0128	3.2E-09	0/1
18 33724354 G T	T	T	rs1941685	ASXL3	0.0531	1.055	0.0094	1.69E-08	0/1
18 43093415 A G	A	G	rs12456492	RIT2	-0.0983	0.906	0.0099	3.8E-23	0/1
18 51157219 T G	T	G	rs8087969	MEX3C	-0.0578	0.944	0.0102	1.41E-08	./
19 2341049 C T	T	C	rs55818311	SPPL2B	-0.0696	0.933	0.0111	4.18E-10	0/0
2 101780501 T A	A	A	rs11683001	MAP4K4	0.0705	1.073	0.0098	8.04E-13	0/1
2 134681219 G A	A	G	rs4954162	TMEM163	-0.0746	0.928	0.0132	0.00000015	0/0
2 134707046 A G	A	A	rs57891859	TMEM163	0.0807	1.084	0.0107	4.55E-14	0/1
2 17966582 A T	A	A	rs76116224	KCNS3	0.1104	1.117	0.0194	1.27E-08	0/0
2 95335195 A T	A	T	rs2042477	KCNIP3	-0.0657	0.936	0.0116	1.38E-08	0/1
20 3184040 T C	T	T	rs2295545	DDRKG1	0.0622	1.064	0.0096	8.45E-11	0/1
3 151391177 T A	A	T	rs11707416	MED12L	-0.0627	0.939	0.0097	1.13E-10	0/1
3 161359842 A G	A	G	rs1450522	SPTSSB	-0.0616	0.940	0.0099	5.01E-10	0/1
3 183042285 T G	T	T	rs10513789	MCCC1	0.1485	1.160	0.0121	1.22E-34	0/1
3 28664199 T C	T	T	rs6808178	LINC00693	0.0658	1.068	0.0096	8.09E-12	0/1
4 169662006 G A	A	G	rs62333164	CLCN3	-0.0638	0.938	0.01	2E-10	0/0
4 17967188 T A	A	T	rs34025766	LCORL	-0.0839	0.920	0.0133	2.87E-10	0/0
4 76189212 C G	C	C	rs6825004	SCARB2	0.0622	1.064	0.0102	1.17E-09	./
4 76226816 A G	A	G	rs4101061	FAM47E	-0.0912	0.913	0.0102	4.97E-19	0/1
4 76276901 C T	T	C	rs6854006	FAM47E-STBD1	-0.0912	0.913	0.0097	5.82E-21	0/1
4 89685975 C G	C	C	rs356228	SNCA	0.1503	1.162	0.0102	9.53E-49	0/0
4 89704960 G A	A	G	rs356182	SNCA	-0.2774	0.758	0.0105	3.89E-154	0/1
4 89744890 C T	T	C	rs356203	SNCA	-0.2504	0.778	0.0096	5.16E-149	0/1
5 103030090 G C	C	C	rs26431	PAM	0.0621	1.064	0.0103	1.57E-09	1/1
5 134863415 C A	A	C	rs11950533	C5orf24	-0.0916	0.912	0.0158	7.16E-09	0/0
6 111922088 A G	A	A	rs997368	FYN	0.0714	1.074	0.0119	1.84E-09	0/1
6 71778059 C T	T	T	rs12528068	RIMS1	0.0657	1.068	0.0103	1.63E-10	0/1
7 23260430 A C	A	A	rs199351	GPNMB	0.1016	1.107	0.0096	5.25E-26	0/1
7 66544864 T A	A	T	rs76949143	GS1-124K5.11	-0.1432	0.867	0.0253	1.43E-08	0/1
8 11854934 A C	A	A	rs1293298	CTSB	0.093	1.097	0.0114	3.99E-16	0/1
8 129889663 T C	T	C	rs2086641	FAM49B	-0.0605	0.941	0.0107	1.81E-08	0/1
8 16840084 G T	T	G	rs620513	FGF20	-0.0856	0.918	0.0108	2.72E-15	0/1
8 22668467 T C	T	T	rs2280104	BIN3	0.0556	1.057	0.0098	1.16E-08	0/1
9 17579692 T G	T	G	rs13294100	SH3GL2	-0.0859	0.918	0.01	8.72E-18	0/1
9 17727067 A G	A	G	rs10756907	SH3GL2	-0.0926	0.912	0.011	5.06E-17	0/1
9 34046393 C T	T	C	rs6476434	UBAP2	-0.0615	0.940	0.0106	6.58E-09	0/0

3. Page 7: What is meant by “wildtype” iPSC-midbrain neurons, and how is this defined? What is the source of these neurons? How are you defining “human midbrain cultures” and “human midbrain neurons” are you distinguishing between these descriptions in the text?

Response: We apologize for the confusion on this matter. We have updated the text to reflect the controls used, which are isogenic corrected lines that are background matched to the A53T iPScn line (described in comment 2), and the isogenic corrected line that matches SNCA triplication line (SNCA-3x). We originally stated ‘wildtype’, in the first version, referring to the wildtype SNCA sequence present in the isogenic corrected lines. We have extensively characterized the SNCA 3X and isogenic corrected lines previously in Stojkovska et al, Neuron, 2022. The source of the original SNCA 3X line was a lymphoblast line from the Coriell Cell repository. We reprogrammed the line to iPSCs in-house. The identifier codes are included in the Materials and Methods section, under the Key resources table.

Please note that “human midbrain cultures” and “human midbrain neurons” refer to the same thing. Since the cultures are ~90% biii-Tubulin + cells, they are nearly homogenous iPSC-derived neuron cultures. We define the cultures as presence of DAergic markers including LMX1a, FOXA2, Tyrosine Hydroxylase, b-iii-Tubulin and other markers. Additionally, differentiated cells are generated using a standardized protocol developed in the Studer lab (Kriks et al, Nature, 2011), which is widely used in many labs, and we have been using it since 2011 after being trained directly by the Studer lab. iPSC-derived neurons made by this protocol synthesis, store, and release DA, and exhibit many other features are authentic human DA neurons including electrophysiological properties of midbrain neurons (Kriks et al, Nature, 2011). We have extensively characterized PD pathological features from cells made using this protocol (eg, see Zunke et al, Neuron, 2018; Cuddy et al, Neuron, 2019; Fredriksen et al, PNAS 2021; Stokovska et al, Neuron, 2022, and Pitcairn et al, J. Neurosci 2023). We routinely perform quality control analysis as shown above in Reviewer Figure 8.

For clarity, we now defined the cultures used in this study as iPSC-derived midbrain cultures (abbreviated iPScn), and cite the protocol used in the methods section.

4. Regarding mouse tissues that were “previously collected.” Are these mouse tissues from a published study? The origin of the tissues and how they were obtained should be described in more detail. Were wildtype littermates or other mice used as controls? The absence of any IACUC approval or other information for “tissues from the freezer” is concerning. How many mice? Sex? Age? The methods are not sufficient.

*Response: We apologize for not including this information in the first version. These lysate samples were only used for this study, and we have an approved IACUC protocol for this purpose (Protocol: IS00021691 through the Northwestern University IACUC). We have updated this information in the methods section. We have included the n-number and age in the figure legends. The transgenic line is from Jackson labs, line M83, B6;C3-Tg(Prnp-SNCA*A53T)83Vle/J, Strain #:004479) and non-transgenic controls are from the same genetic background and housed under the same conditions as the Tg mice as described in*

the methods. Although there are no sex-linked phenotype differences that we are aware of in this particular line, the data we show represent mice from two females and one male.

5. The targeted metabolomics methods and analyses are very limited and need more detail. Data are provided for only 5 molecules. The analysis must have been for a panel larger than this? or not? Please provide all metabolomic data.

Response: Since this metabolomics study is not discovery based, but instead is testing a specific hypothesis, we only focused on a select metabolites related to glycolysis and the HBP for this paper. In the results section, we clarified this by writing “ We next considered the possibility that reduced N-glycan synthesis was a result of low intracellular glucose or impaired turnover within the initial steps of glycolysis that provide essential HBP substrates (Figure 2A). We performed targeted metabolomics of A53T iPScn to specifically test this hypothesis.”

Some additional metabolites relevant to the HBP pathway were attempted, but when their detection failed in most samples, we did not show them in the paper since the data were not conclusive. We did get additional measurements from metabolites that are not directly related to this pathway, however we have yet to analyze and interpret the findings. We plan to publish that data in a different study as it relates to a distinct discovery-based hypothesis, and including the data here would obscure the focus of our study.

6. Methods: What does “subject” mean the Methods section?

Response: The word ‘Subject’ was included in the subtitle of a standardized Methods template that we use for many of our studies. We now changed it to “Sample” to avoid confusion, which refers to the human brain material used in the study. We also now provide a spreadsheet (Supplemental Data Table 1) that includes the clinical and pathological data from the brain samples used here, and their link to the main figures.

7. Methods: As above, transgenic mouse methods are inadequate.

Response: We apologize for the missing information and we now have included the relevant information including stock number from Jackson labs, controls used, and IACUC protocol number. This information can be found under the Methods section, “Experimental Models and Sample Details”. The way the samples were processed and analyzed can be found under their respective subtitles in the Methods details section. For example, under “Sequential protein extraction and western blotting analysis” second paragraph, the western blot methods can be found. Since the mouse lysates and iPScn lysates are processed and analyzed by similar methods, we referred to the appropriate sections to avoid repetition.

8. Methods: Methods for metabolomics, as above, are inadequate. What was this targeted panel or certain molecules only were chosen? What were the standards? Where are the data?

Response: We apologize for not including this information in the first version. We now include a detailed method for the metabolomics study (shown below for convenience). The data are shown in figure 2 B-F, normalized to total protein, then expressed as fold change to facilitate the visualization of the changes between Corr and A53T iPSn. We requested a targeted metabolomics analysis for glycolysis and the HBP from the MIT metabolomics core given that our study is hypothesis based (as mentioned in comment # 5, rev 3 above). We used isotopically labeled glutamate-13C5-15N as an internal standard. Since we know the concentration of this standard when we put it in the extraction buffer, this serves as a way to determine if metabolite degradation or break-down occurs as a result of the extraction process or storage. It can also determine if there is a problem (ie, needle clog) or inaccuracies with the sample injector volumes on the UPLC. We typically do not publish the data from the internal standards since these quantification methods are part of the standard operating procedures, but we provide the raw peak intensity values below as Reviewer Figure 12, representing data from individual culture well replicates. Note that the metabolomics core had identified one outlier that was excluded from our analysis in the Corr set because it failed the QC testing.

Methods section:

“iPSn in a 6 well format were placed on ice and quickly washed with ice-cold 0.9% NaCl. The plates were then placed on dry ice, followed by addition of 1 ml/well of 80% methanol extraction mix, containing isotopically labeled internal standard, Glutamate 13C5-15N (Cambridge Isotope Laboratories). Cells were thoroughly scrapped and transferred to pre-chilled Eppendorf tubes. The tubes were vortexed for 10 min at 4°C. The samples were spun down at 21,000 X g for 10 min at 4°C, and the supernatant was speedvac dried and shipped on dry ice to the Metabolite Profiling Core Facility at the Whitehead Institute of the

Massachusetts Institute of Technology (<https://biology.mit.edu/tile/metabolite-profiling-core-facility>). The pellet was resolubilized in 2% SDS lysis buffer and subjected protein assay by DC™ Protein Assay Kit (#500, Bio-Rad). The data was normalized to total protein. Relative abundance of each metabolite was obtained by UPLC-MS analysis using a Dionex UltiMate 3000 ultra-high performance liquid chromatography (UPLC) system using hydrophilic interaction chromatography (HILIC), and coupled to a Thermo Scientific QExactive Orbitrap mass spectrometer. Metabolite measurements were calculated by dividing the Raw peak area of each metabolite with the peak area of the internal standard – one of the isotopically labeled amino acids included in the extraction mix, which was then normalized to protein. For quality control (QC) analysis, a “pool” of sample consisting of a mixture of several uL from each of the biological samples was run, which created a representative sample that was run multiple times to get a measure of technical reproducibility for each metabolite. A CV (standard deviation / average) was calculated for these technical replicates and metabolites with a CV < 0.25 were considered reliably detected. Also, a larger injection volume was run with 0.3-fold and 0.1-fold dilutions of the pooled sample, which indicated whether samples fall in the linear range of detection for each metabolite, or there was a detector saturation or nearing the lower limit of detection. Metabolites that had a linear correlation coefficient $R < 0.975$ for this dilution series were flagged.”

9. References are not in proper format.

Response: We thank the reviewer for noting this. We have double checked the formatting, and according to our EndNote software, the citations are in the standard Nature style.

Figures:

Response: For all of the comments below, we now provide full blot images that were questioned by the reviewer, and all blots that appear in the paper will be available as a downloadable link if the paper should be accepted for publication. We thank the reviewer for bringing up these points and we have now double checked all of our blots that appear in the paper. We confirm that all of our loading controls are derived from the same exact membrane, as we re-probe the membrane sequentially for a loading control (GAPDH or β -Tubulin) after each protein of interest is detected. We also use Coomassie brilliant blue staining of the same gel that was used for membrane transfer. The gels left over after the membrane transfer always have proteins left on the gel to visualize and accurately quantify. Please note that it is very difficult to determine if proteins of different molecular weights are from different or same blots just by looking at band shapes. In our experience, the bands shapes can change (eg smiling or frowning or wavy) depending on the vertical position on the blot. For example, sometimes low MW proteins frown, while high MW proteins from the same lane appear straight and vice versa. This is likely due to slight differences in the ionic compositions of the samples or gels and that the electrical resistance changes during the

electrophoresis run. We provide examples of this below, and show the full blot images of each figure panel mentioned.

Figure 3B: The GAPDH blot (first blot, top to bottom) looks like it is from a different experiment than the other 3 blots.

Response: We apologize for this, as it should have been mentioned in the figure legend that Figure 3B shows data from two different membranes, which is why GAPDH was originally shown twice (to match each individual membrane). However, the lysates used are from the same experiment (using the same lysate, but loaded on two different gels). We now write in the legend “GFPT2 and GRP78 are from different blots, and GAPDH loading control shown below corresponds to the same matching blot.”

Our western blot system uses two channels that can simultaneously detect primary antibodies from rabbit or mouse, when the secondaries are conjugated to two different fluorophores. For example, we use anti-mouse Alexa 680 together with anti-rabbit IRDye 800. In the blot below, we used IRDye800 to detect GFPT2 (rabbit antibody), and on the same membrane, have sequentially probed for GAPDH (mouse antibody), using the anti-mouse Alexa-680 conjugate on a different channel that does not cross react with the other channel. The red boxes indicate the portions used for figure 3B. Note that additional bands appear because we use the membrane as much as possible for additional markers in case we need them (for example, the biii-tubulin band (smiling, fat band) can be seen above GAPDH band at about 50kDa).

Additionally, we have increased the size of representative bands from the same lane to demonstrate to the reviewer that bands of different MWs within the same gel lane can have different shapes. In the A53T + Tunic lane, the shapes change from top (high MW, slight frown) to bottom (low MW slight smile, and straight). For this reason, it is not possible to determine if bands are from different or same blots by examining their shape.

Figure 4A: The GFP blot does not appear to be the same blot.

Response: Note that this is now Figure 5A. We thank the reviewer for bringing this to our attention. We have double checked this, and can confirm that the GFP blot is in fact from the same blot as GFPT2 and GAPDH signal. For the reasons mentioned above, it is not possible to determine that the signals from different proteins are from different blots by their appearance alone. We show the full blots below (which will also be available for download if the paper is accepted). The red boxes indicate the portions used in Figure 5A. The black dotted line indicates the region of the lower blot that was increased by 300% so that the different band shapes can be more easily seen. Even though they are from the same vertical gel lane, they have different shapes.

Figure 4D: Are these blots all from the same experiment?

Response: Please note that this is now Figure 5E. We confirm that these are all from the same experiment. The signals for immature and mature HexB is not always equally intense, so we have to use two different intensities in this figure to avoid saturation of the immature bands. For example, when the 64 and 50 kDa MW bands are visible at the appropriate exposure, the mature bands become saturated and it is not possible to see any differences (left blot). However, reducing the intensity, we can now see the differences in the mature forms, but the immature forms disappear (middle blot). We apologize for omitting this information. We now have noted this in the figure legend. We now write: “Note that the Immature Hex B bands are shown at a higher intensity compared to the mature Hex B bands, to avoid saturation of the mature forms.”

Figure 5C: No Veh is indicated. What is the vehicle?

Response: Note that this is now Figure 6C. We apologize for omitting this information. The vehicle in each panel shown in PBS. We now have updated this information in the figure legends and methods.

Figure S2A: Is GAPDH from the same blot?

Response: We can confirm that the GAPDH is from the same blot. In the first version, the HK-1 blot was exported at a slight angle by mistake, while the GAPDH was straight, and we now straightened both of the signals from the same blot in the Figure. Full blots are shown below.

Figure S3D: The blot for a-syn is rather curved and not like the other blots cut out here. Please explain.

Response: Note that this is now Figure S5C. We apologize for not explaining this in the figure legend and have now updated it. Both GFPT2 and a-syn have their corresponding loading controls shown beneath them and were run on the same gel. The top blots showing GFPT2 and GAPDH are matched and from the same exact lanes on the blot (full blots shown below on the left side). The bottom blots showing a-syn and GAPDH are also matched and from the same exact lanes on the blot (shown on the right side of the figure below). As mentioned above, the fact that a-syn is curved but the GAPDH is not is from the fact that the band shapes are not always the same within one gel lane. You can see that other bands above a-syn at 36 kDa and higher appear straight, while a-syn is curved down. The left blots show

GFPT2 and GAPDH, which correspond to the blots shown in figure S5C on the top, boxed in red. The right blots show α -syn and GAPDH, which correspond to the blots shown in figure S5C on the bottom, boxed in red.

Figures S5C

Comments below are related to clarity of the text:

1. Page 2, bottom: Please explain, “express wildtype GBA1 mutations.” Do you mean to say ..”do not express pathogenic GBA1 variants.”?

Response: We apologize for this error. It was meant to say ‘wildtype GBA1’, and we have now corrected this in the text.

2. Throughout the manuscript: Since the terminology “A53T” is historical, I suggest using approved human genome nomenclature for all variants – at least at the beginning and in the methods so that there is no confusion and all readers are included.

Response: We have now included “SNCA G209A (A53T α -syn)” in the beginning of the methods section to clarify this.

3. Throughout the manuscript: Replace “mutation” with “variant” or “pathogenic variant” depending on context. This is the currently accepted terminology.

Response: We now replace mutation with “pathogenic variant” to refer to the mutations A53T a-syn, SNCA 3x a-syn, and L444P GCase.

4. Throughout the manuscript, please be consistent: (a) in terminology: a-syn or alpha-syn or α -syn. Choose one and use it. (b) with all abbreviations: hours, h; minutes, min, etc. (c) in all punctuation and spacing: i.e. 10mM or 10 mM; n=3 or n = 3; p<0.005 or p < 0.005, etc. (d) define all abbreviations at first use

Response: We thank the reviewer for this suggestion and have now corrected these errors. We use a-syn throughout, h for hours, min for minutes, and s for seconds (we defined the abbreviations in the first text appearance). We have put a space between the number and units used (eg, 10_mM).

REVIEWER COMMENTS

Reviewer #1 (Remarks to the Author):

The revised paper by Wani et al. addresses several of our previous points and provides additional new results. However, the precise mechanisms underlying the effect of enhancing the hexosamine biosynthetic pathway in improving the folding of various lysosomal hydrolases remain largely unclear.

Another concern is how to interpret the data in terms of the specificity of the pathway. To address this issue, the authors analyzed cortical postmortem brain lysates from different neurodegenerative diseases. They found that only DLB samples showed reduced N-glycosylated proteins. The authors conclude that "the hypo-N-glycosylation phenotype does not occur in general neurodegeneration, but is specific to synucleinopathies. Recent literature shows alterations in N-glycosylation in other NDDs (e.g. PMID: 33008897, 21629267). Indeed, rather than a global reduction in N-glycosylation, specific protein N-glycosylation patterns (as assessed by more complex glycoproteomic approaches) may be altered in aging and disease. Rather than a unique link between reduced N-glycosylation and synucleinopathies, the data presented by the authors may suggest a unique role for lysosomal function in PD. Would improving lysosomal function via the HBP pathway improve other NDD phenotypes? These points should at least be discussed.

Additional points are listed below:

The authors stated, "We quantified the levels of ER proteostasis machinery (mRNA and protein) after HBP was enhanced by GlcNAc or GFPT2 expression". However, only mRNA levels were examined (Figure 5 and Supplementary Figure 5A). Protein levels should be provided. Furthermore, the data show that the effects on lysosomal hydrolases do not occur through increased N-glycosylation, as opposed to indirect effects mediated by ER chaperones alone. This is in contrast to the original work by Denzel et al. While differences in model organisms should be acknowledged, such a discrepancy should be discussed.

Reviewer #2 (Remarks to the Author):

I thought this was a good paper in the initial submission, and the authors have further improved the manuscript in the revision. This should be published in Nat Comm.

Reviewer #3 (Remarks to the Author):

I appreciate the efforts to address the points in the initial reviews. The authors have adequately

addressed the concerns and have further supported their findings.

Minor comments regarding the supplemental tables:

1. Define PMI
2. Is Sex unknown for CNTRL 6 (case #15)? If so, please denote this in the table. If not, then please make a clear indication for of N/A or unknown. Please this for each instance
3. "Mutation" is not the current term—suggest changing column header to Genomic Variant or Genomic Findings

Response to Reviewers

We thank the reviewers for carefully going through the new data in the revised manuscript. In this 3rd revision, we added three additional experiments to further support the mechanistic link between the HBP enhancement, hydrolase activity, and reduction of protein aggregates (requested by Reviewer 1). We also expanded the discussion section to include connections between our data and N-glycosylation in other diseases, as well as to that of Denzel et al Cell 2014, and corrected the supplemental table as suggested by Reviewer 3. We respond below in a point-by-point manner.

Reviewer #1 (Remarks to the Author):

1) The revised paper by Wani et al. addresses several of our previous points and provides additional new results. However, the precise mechanisms underlying the effect of enhancing the hexosamine biosynthetic pathway in improving the folding of various lysosomal hydrolases remain largely unclear.

*Response: To address this concern, we performed a few additional rescue experiments to support our conclusions including measurement of CANX binding activity and N-glycosylated levels of GCase after GFPT2 overexpression, shown in Figure 5D and Supplemental Figure S5B-C. The rescue of lysosomal function by GFPT2 restoration or GlcNAc occurs through **1) Increasing N-glycosylation of lysosomal enzymes (Shown in Figure 5C, 6A, and New Figure 5D); 2) This in turn Increases binding activity of ER chaperones including Calnexin, which is a known GCase chaperone (New Figure 5D, see increased binding of Calnexin with N-glycans), 3) Improved chaperone activity promotes properly folded, mature GCase and Hex-B (Figure 5E, F; Figure 6B-F); 4) Lysosomal targeting is improved since we show increased EndoH resistant forms of GCase and Hex, increased activity within lysosomal compartments and reduction of a-synuclein (Figures 5I; S5B, C; Figure 6D); 4)***

GFPT2 increases lysosomal function by enhancing N-glycosylation of hydrolases, and not by upregulating other UPR chaperones (Supplemental Figure S5A, B and S6A). We added mRNA levels of Calnexin to support these findings (New Figure S5A, right graph). We also know that GFPT2 increases GCase/Hex-B enzymes that are properly folded, since the enzymes are active and transported to lysosomes (Figure 5F, G, and I; Figure 6D) (only properly folded GCase are permitted to exit the ER). We summarized these results in Figure 7.

2) Another concern is how to interpret the data in terms of the specificity of the pathway. To address this issue, the authors analyzed cortical postmortem brain lysates from different neurodegenerative diseases. They found that only DLB samples showed reduced N-glycosylated proteins. The authors conclude that "the hypo-N-glycosylation phenotype does not occur in general neurodegeneration, but is specific to synucleinopathies. Recent literature shows alterations in N-glycosylation in other NDDs (e.g. PMID: 33008897, 21629267). Indeed, rather than a global reduction in N-glycosylation, specific protein N-

glycosylation patterns (as assessed by more complex glycoproteomic approaches) may be altered in aging and disease. Rather than a unique link between reduced N-glycosylation and synucleinopathies, the data presented by the authors may suggest a unique role for lysosomal function in PD. Would improving lysosomal function via the HBP pathway improve other NDD phenotypes? These points should at least be discussed.

Response: We agree and have now re-worded the conclusions in the results section to read “We only found a significant reduction in N-glycosylated proteins of DLB brains compared to control, although mean values for other disease groups were also reduced but without significance (Figure 1C).”

We also cite the papers listed above in the discussion section, last paragraph : We showed that global N-glycosylation reduction occurs in DLB brain using Con-A reactivity, while others have shown more complex changes in N-glycan patterns using different techniques in different neurodegenerative diseases ^{52,53}. This indicates that n-glycosylation changes may play a role in the pathogenesis of multiple diseases.”

Lysosomal dysfunction has been documented in multiple neurodegenerative diseases including nearly all lysosomal storage disorders, and genetic links have been shown in PD, DLB, FTD. It is possible that any NDD with protein accumulation will benefit from enhanced lysosomal function, by decreasing protein aggregates. However, at this point we cannot speculate if enhancing the HBP would translate to increased lysosomal function in other diseases where there is no change in the HBP. For example, we were not able to increase N-glycosylated proteins in isogenic corrected cells that have normal levels of GFPT2 by GlcNAc supplementation (Figure 6A). Also, other diseases may have perturbations in pathways downstream from ER proteostasis, and therefore boosting ER proteostasis may not provide any benefit. However, work from Denzel et al, Cell, 2014, suggests that enhancing the HBP can in fact reduce many different protein aggregates including a-synuclein, a-beta and expanded poly-Q. Future studies will determine if increasing the HBP can reduce protein aggregates in iPSC models of other diseases, but our seminal work in synucleinopathies lay the groundwork for such studies.

Additional points are listed below:

3) The authors stated, "We quantified the levels of ER proteostasis machinery (mRNA and protein) after HBP was enhanced by GlcNAc or GFPT2 expression". However, only mRNA levels were examined (Figure 5 and Supplementary Figure 5A). Protein levels should be provided. Furthermore, the data show that the effects on lysosomal hydrolases do not occur through increased N-glycosylation, as opposed to indirect effects mediated by ER chaperones alone. This is in contrast to the original work by Denzel et al. While differences in model organisms should be acknowledged, such a discrepancy should be discussed.

Response: We agree this topic requires more discussion and now included an extra

paragraph in the discussion (2nd to last paragraph). We now provide protein levels of GRP94, GPRP78, and Calnexin, along with the mRNA and protein levels in Supplemental figure 5A and B. Also, Supplemental figure 6A includes protein analysis by western blot. Our data indicates that neither mRNA or protein of GRP78 and GRP94 are increased when the HBP is enhanced by either GFPT2 overexpression or GlcNAc. GRP78 and GRP94 are two sensitive and reliable indicators of UPR and general ER proteostasis. Since the levels of the ER chaperones are not changed by HBP enhancement, the effects we observe upon GFPT2 overexpression cannot occur from upregulation of ER chaperones. We therefore concluded that the rescue of lysosomal function occurs through increased N-glycosylation (see response to #1 above). In general, our findings are consistent with Denzel et al.

In Denzel et al, Cell 2014, the authors showed that an aggregation prone reporter fused to YFP was decreased in GFAT-1 GOF mutants. Consistent with our work, the authors also concluded that N-glycosylation was required for the reduction of aggregates. In their paper, they write : “Inhibiting N-glycosylation by knockdown of oligosaccharyltransferase (OST) complex components suppressed the reduction of SRP-2H302R::YFP puncta and lifespan extension found in *gfat-1* *gof* mutants (Figures S4A and S4B). These data suggest that N-glycosylation is required for improved protein homeostasis and longevity in *gfat-1* *gof* mutants”.

The KD of the IRE-1 pathway abolished the protective effect gained by the GFAT GOF mutant, presumably because the GFAT-1 mutant was not expressed at high enough levels (given that XBP1s controls GFAT1 expression, Wang et al, Cell, 2014). However this dependence of *gfat1* on the IRE-1 pathway is upstream of GRP78 and GRP94 expression. The authors also analyzed the levels downstream UPR components and write “Unexpectedly, however, we found no changes in transcriptional regulation of known UPR target genes by qPCR (Figure 4C), including *hsp-4*, *spliced xbp-1*, and target genes of the ATF-6 and PEK-1 UPR branches (Shen et al., 2005). HSP-4 protein levels were slightly increased without reaching statistical significance, and CNX-1 protein levels were unchanged (Figures S4C and S4D), suggesting that a functional UPR is required but not activated in *gfat-1* *gof* mutants.”

Importantly, *C. elegans* HSP-4 is the GRP78 homolog in humans, and its mRNA was not changed in Denzel et al (Fig. 4C of that paper), consistent with our data in Supplemental Figure S5A.

Overall, we feel our data are consistent with that of Denzel et al, since we show that lysosomal activity is enhanced by the HBP. Denzel et al showed that autophagy was increased by GFAT-1 GOF mutants, and since autophagy requires lysosomes for clearance, their data indicate a similar effect on proteostasis. Although we did not specifically test the delivery of cargo through autophagy in our paper, future work will address this in PD iPSn.

Reviewer #2 (Remarks to the Author):

I thought this was a good paper in the initial submission, and the authors have further

improved the manuscript in the revision. This should be published in Nat Comm.

Response: We thank the reviewer for the encouraging comments.

Reviewer #3 (Remarks to the Author):

I appreciate the efforts to address the points in the initial reviews. The authors have adequately addressed the concerns and have further supported their findings.

Minor comments regarding the supplemental tables:

1. Define PMI

Responses: Thank you for picking up this error. We have now defined PMI – post-mortem interval

2. Is Sex unknown for CNTRL 6 (case #15)? If so, please denote this in the table. If not, then please make a clear indication for of N/A or unknown. Please this for each instance

Response: We now have updated the information in the excel sheet – control 6 is Male.

3. “Mutation” is not the current term—suggest changing column header to Genomic Variant or Genomic Findings

Response: Thank you for this suggestion. We now have updated this heading in the excel sheet.

REVIEWERS' COMMENTS

Reviewer #1 (Remarks to the Author):

The authors have thoroughly addressed all of our previous concerns.